# Exploiting the kinesin-1 molecular motor to generate a virus membrane penetration site

Madhu Sudhan Ravindran[1], Martin F. Engelke[1], Kristen J. Verhey[1] & Billy Tsai[1]

Viruses exploit cellular machineries to penetrate a host membrane and cause infection, a process that remains enigmatic for non-enveloped viruses. Here we probe how the non-enveloped polyomavirus SV40 penetrates the endoplasmic reticulum (ER) membrane to reach the cytosol, a crucial infection step. We find that the microtubule-based motor kinesin-1 is recruited to the ER membrane by binding to the transmembrane J-protein B14. Strikingly, this motor facilitates SV40 ER-to-cytosol transport by constructing a penetration site on the ER membrane called a 'focus'. Neither kinesin-2, kinesin-3 nor kinesin-5 promotes foci formation or infection. The specific use of kinesin-1 is due to its unique ability to select post-translationally modified microtubules for cargo transport and thereby spatially restrict focus formation to the perinucleus. These findings support the idea of a 'tubulin code' for motor-dependent trafficking and establish a distinct kinesin-1 function in which a motor is exploited to create a viral membrane penetration site.

[1] Department of Cell and Developmental Biology, University of Michigan Medical School, 109 Zina Pitcher Place, 3043 BSRB, Ann Arbor, Michigan 48109, USA. Correspondence and requests for materials should be addressed to M.S.R. (email: mravindr@umich.edu) or to B.T. (email: btsai@umich.edu).

Viruses co-opt cellular machineries to cause infection. An essential infection step is virus penetration across a host membrane, to gain access into the cytosol. For enveloped viruses, fusion between the viral and a host membrane delivers the core viral particle into the cytosol[1,2]. By contrast, the molecular mechanism driving membrane penetration of a non-enveloped virus remains poorly understood[3]. In particular, a key question is whether the virus passively exploits a pre-existing protein-conducting channel to cross a membrane or actively remodels a membrane's property to promote its translocation across the lipid bilayer. Indeed, membrane translocation of the non-enveloped polyomavirus (PyV) highlights this enigma.

PyVs are responsible for many debilitating human diseases, especially in immunocompromised individuals. Prominent human PyVs include the BK PyV that induces haemorrhagic cystitis and nephropathy, JC PyV that triggers progressive multifocal leukoencephalopathy and the Merkel cell PyV that causes Merkel cell carcinoma[4]. Simian virus 40 (SV40) represents the archetype PyV, possessing not only structural and genetic similarities to human PyVs, but also shares the same infection pathway as its human counterparts[4]. Not surprisingly, studies on SV40 entry have historically illuminated the cellular basis of human PyV infection. SV40 consists of 72 pentamers of the structural protein VP1 that encases its DNA genome, with each pentamer harbouring an internal hydrophobic protein VP2 or VP3. When properly assembled, the viral particle displays a diameter of $\sim 45$ nm[5,6]. To infect cells, SV40 binds to the ganglioside GM1 receptor on the plasma membrane, is endocytosed and targets to endolysosomes[7–9]. The virus then sorts to the endoplasmic reticulum (ER) where it penetrates the ER membrane to access the cytosol[10–13]. In the cytosol, SV40 traffics to the nucleus where transcription and replication of the viral genome lead to lytic infection or cellular transformation[14]. Although the molecular basis by which this non-enveloped virus penetrates the ER membrane, a decisive infection step, remains largely mysterious, aspects of this process are being revealed.

SV40 was initially proposed to hijack a cellular quality control pathway called ER-associated degradation (ERAD) to reach the cytosol[12]. During ERAD, a misfolded ER protein is translocated across a protein-conducting channel to reach the cytosol where the misfolded client is degraded by the proteasome[15,16]. However, as SV40 penetrates the ER membrane as a relatively large (45 nm) particle that is unlikely to thread through the pore of a typical channel, a different model describing its membrane transport has emerged[17,18].

In this alternative model, incoming SV40 is hypothesized to remodel the ER membrane to create a membrane penetration site. Consistent with this, SV40 was found to reorganize select ER membrane proteins into discrete puncta called 'foci' where the viral particles enter the cytosol[18]. For example, the transmembrane proteins B-cell receptor-associated protein 31 (BAP31) and BAP29 mobilize into the foci during infection[18]; these membrane factors serve as sensors to detect membrane-embedded SV40 and initiate the membrane translocation event[18]. Likewise, during SV40 infection, the transmembrane J-proteins DNAJ homologue subfamily B member 14 (B14), B12 and C18 accumulate in the foci where they recruit the cytosolic chaperone complex composed of heat shock cognate protein 70 (Hsc70), small glutamine-rich tetratricopeptide repeat-containing protein-α (SGTA) and heat shock protein 105 (Hsp105); this complex extracts SV40 into the cytosol to complete the membrane translocation process[19–21]. Although increasing evidence supports the notion that virus-induced foci function as cytosol entry site during SV40 ER membrane transport[21], how this sub-organellar structure is constructed remains completely unknown. Indeed, what cellular mechanisms are hijacked to accomplish this feat?

Using a combination of biochemical, cell-based and microscopy approaches coupled with a chemical-induced dimerization strategy, our results demonstrate that the force generated by the kinesin-1 molecular motor is harnessed to promote foci formation during SV40 cytosol entry. Although this molecular motor is known to regulate viral intracellular trafficking and disassembly[22–24], our report here ascribes a distinct function to kinesin-1 in which the ability of this motor to read the 'tubulin code' is exploited to promote the localized construction of virus membrane penetration site.

## Results

**Kinesin-1 promotes SV40 infection.** An RNA interference screen identified B14 as an ER membrane J-protein required for SV40 membrane penetration[11]. Using a HEK 293T cell line (293T-REx) stably expressing 3x-FLAG tagged B14 (B14$^{3xFLAG}$), we immunoprecipitated B14$^{3xFLAG}$ and subjected the precipitated sample to mass spectrometry analysis. This pinpointed the cytosolic Hsc70-SGTA-Hsp105 chaperone complex as B14-interacting partners that extract SV40 into the cytosol[11,19,21]. As the kinesin family proteins kinesin-1 (KIF5B and KIF5A) and kinesin-5 (KIF11) were also found in the same precipitated material[21], we asked whether they also play a role during SV40 infection, in particular during the ER membrane penetration phase.

As kinesin motors rely on microtubules to transport cargos, we first tested whether an intact microtubule system is required during SV40 infection after the virus has reached the ER from the cell surface. Simian CV-1 cells are the normal permissive cell line used to study SV40 infection. Accordingly, CV-1 cells were infected with SV40 and at 5 hpi (hours post infection), treated with the microtubule depolymerizing agent nocodazole. We chose this post-infection time point to add the drug, because a majority of SV40 reaches the ER from the plasma membrane $\sim 5$–6 hpi[12,17] (see also Supplementary Fig. 1). Therefore, any effect nocodazole exerts on virus infection should reflect a requirement of microtubules post ER arrival. Using this protocol, we found that nocodazole blocked expression of large T antigen (TAg) in a concentration-dependent manner (Fig. 1a, top panel); TAg is a virally encoded protein expressed only after the virus reaches the host nucleus and is the earliest marker of successful infection. These data suggest that microtubules are important during SV40 infection at a step after virus arrival to the ER from the cell surface, raising the possibility that motor proteins execute a function at this later entry stage. If nocodazole was simultaneously added at infection, virus arrival to the ER was blocked[17], indicating that an intact microtubule network also controls SV40 trafficking from the plasma membrane to the ER.

To investigate whether the kinesin-5 family motor KIF11 facilitates SV40 infection, we immunostained for TAg expression in infected cells treated or not with the highly specific KIF11 inhibitor S-Trityl-L-cysteine (STLC)[25–27]. Representative images of TAg in infected and uninfected CV-1 cells are shown in Fig. 1b. Importantly, when STLC was added 5 hpi, SV40 infection was unaffected, whereas addition of nocodazole (1 μM) or the proteasome inhibitor MG132 (20 μM) at 5 hpi potently blocked infection as expected (Fig. 1c,d); MG132 was previously shown to impair SV40 ER-to-cytosol transport[17]. These results indicate that kinesin-5 does not regulate SV40 entry into the cytosol from the ER to facilitate infection.

We next examined the role of kinesin-1 in SV40 infection by using a dominant-negative (DN) overexpression approach. To this end, we used a motor-less version of wild-type (WT) kinesin-1 motor KIF5B that contains the cargo-binding tail domain and acts in a DN manner (KIF5 DN) to suppress endogenous kinesin-

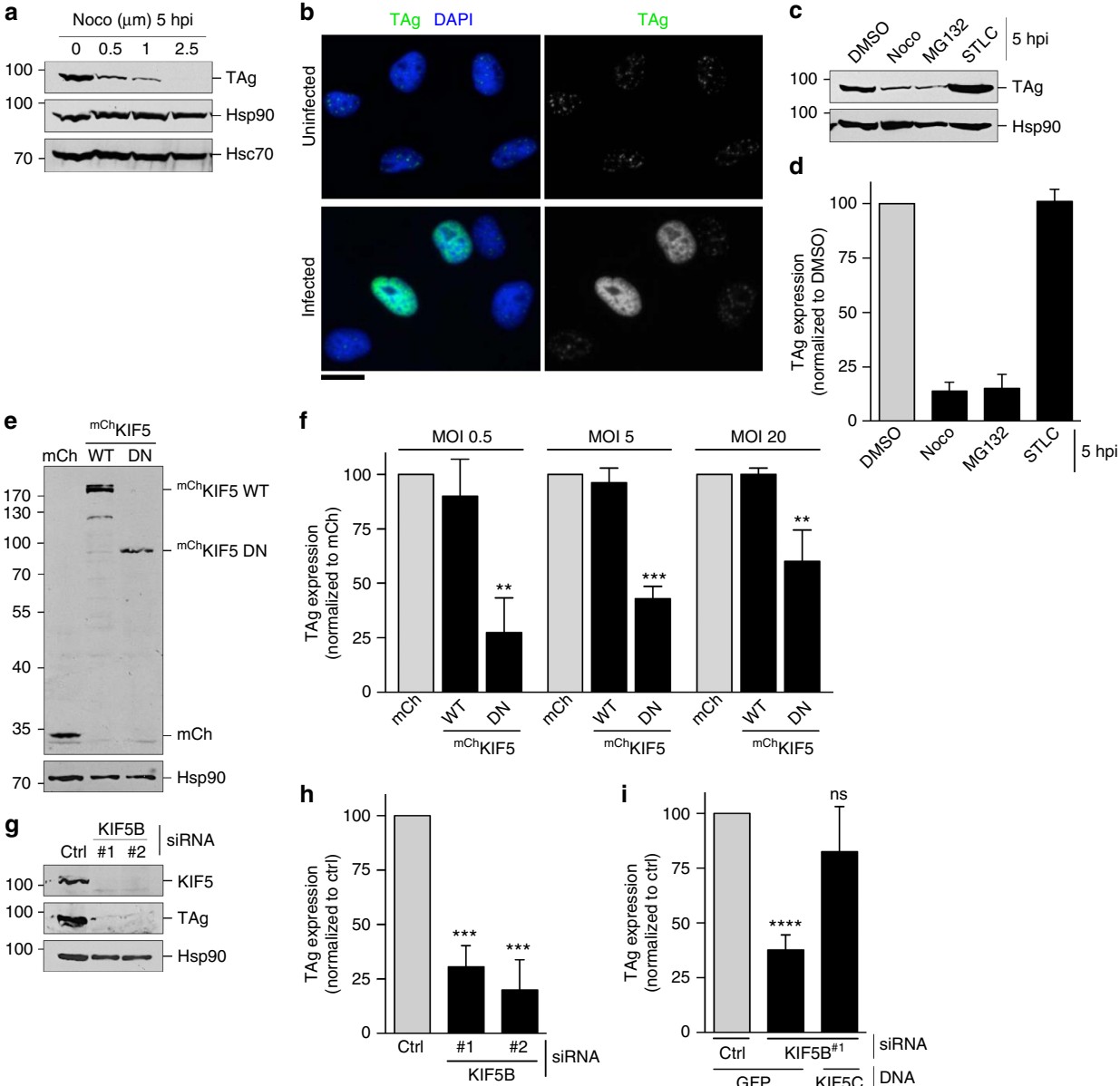

**Figure 1 | Kinesin-1 promotes SV40 infection.** (**a**) CV-1 cells were infected with SV40 (MOI 0.5) at 5 hpi, treated with different concentrations (0, 0.5, 1 and 2.5 μM) of nocodazole. Cells (48 hpi) were lysed and the resulting whole-cell lysate was immunoblotted with the indicated antibodies. (**b**) CV-1 cells infected with SV40 for 24 h were fixed and immunostained against TAg. Images were captured by immunofluorescence microscopy. Scale bar, 20 μm. (**c**) CV-1 cells (5 hpi) were treated with different drugs (1 μM nocodazole, 20 μM MG132 or 1 μM STLC). Cell lysate (48 hpi) was immunoblotted with the indicated antibodies. (**d**) As in **c**, except 24 hpi, infection was scored by immunofluorescence microscopy. Data are normalized to the DMSO control (grey bar). Values represent average of the mean ($n = 3$) ± s.d. (**e**) CV-1 cells expressing the indicated constructs were immunoblotted using a mCh or Hsp90 antibody. (**f**) Cells in **e** were infected with SV40 (MOI 0.5, 5 and 20) for 24 h, fixed and immunostained against TAg. Cells expressing mCh-tagged proteins were scored for TAg positive signal by immunofluorescence microscopy. Data are normalized to the mCh control (grey bar). Values represent average of the mean ($n = 3$) ± s.d. *P*-values were calculated with the two-tailed Student's *t*-test; **$P < 0.01$ and ***$P < 0.001$. (**g**) CV-1 cells were transfected with a control (ctrl), KIF5B[#1] or KIF5B[#2] siRNA. After 48 hpi, the resulting whole-cell lysate was immunoblotted with the indicated antibodies. (**h**) Cells in **g** were infected with SV40 for 24 h, fixed, immunostained against TAg and infection was scored as in **d**. Data are normalized to the ctrl siRNA (grey bar). Values represent average of the mean ($n = 3$) ± s.d. *P*-values were calculated with the two-tailed Student's *t*-test; ***$P < 0.001$. (**i**) Cells transfected with ctrl or KIF5B[#1] siRNA were co-transfected with GFP or a construct simultaneously expressing KIF5C and GFP. Cells were infected with SV40 and TAg expression in GFP-expressing cells were scored as in **f**. Values represent average of the mean ($n = 5$) ± s.d. *P*-values were calculated with the two-tailed Student's *t*-test; ****$P < 0.0001$; ns, not significant.

1 activity[28,29]. This construct was tagged with mCherry (mCh) at its amino terminus, generating [mCh]KIF5 DN. Full-length WT KIF5B tagged with mCh at its N terminus ([mCh]KIF5 WT) and mCh were used as controls. Expression of these proteins was examined by subjecting CV-1 cell lysates to SDS–polyacrylamide

gel electrophoresis (SDS–PAGE) followed by immunoblotting against a mCh antibody (Fig. 1e). When TAg expression was scored in cells expressing the mCh-tagged proteins, we found that overexpression of [mCh]KIF5 DN, but not [mCh]KIF5 WT, significantly impaired SV40 infection when compared with cells

expressing mCh, regardless of the multiplicity of infection (MOI) (0.5, 5 and 20) used in the experiment (Fig. 1f). Hereon, all infection assays were performed at MOI 0.5.

Using an independent approach, we depleted KIF5B in CV-1 cells using two different small interfering RNAs (siRNAs) (KIF5B[#1] and KIF5B[#2]) (Fig. 1g, first panel). When the TAg level was analysed by immunoblotting, we found that depleting KIF5B robustly blocked SV40 infection (Fig. 1g, second panel), similar to results analysed by immunofluorescence (Fig. 1h). To demonstrate that the KIF5B siRNA is not exerting off-target effects, we performed a rescue experiment by expressing the KIF5 family member KIF5C in KIF5B[#1] siRNA-transfected cells. The plasmid that drives the expression of KIF5C additionally drives the expression of green fluorescent protein (GFP), allowing the detection of transfected cells. Cells were then infected with SV40, fixed and scored for the presence of TAg only in GFP-positive cells. Our results demonstrated that expressing KIF5C restored SV40 infection in KIF5B[#1] knockdown cells (Fig. 1i), unambiguously establishing that the block in SV40 infection due to transfecting the KIF5B siRNA results from KIF5B depletion. These collective findings strongly support the importance of kinesin-1 during SV40 infection.

**Kinesin-1 complexes with the ER membrane J-protein B14.** Because of kinesin-1's role in SV40 infection, we assessed the putative interaction between kinesin-1 and the transmembrane J-protein B14, as initially suggested by our mass spectrometry analysis[21]. 293T-REx cells stably expressing B14[3xFLAG] and control 293T cells were transiently transfected with [mCh]KIF5 WT. Lysates derived from both cells were subjected to immunoprecipitation using FLAG-conjugated agarose beads. We found that [mCh]KIF5 WT was pulled down only when B14[3xFLAG] was precipitated from 293T REx but not 293T cells (Fig. 2a top panel, also see Supplementary Fig. 5 for uncropped western blottings), demonstrating that KIF5 interacts with B14. In addition, when FLAG-tagged B14 ([FLAG]B14) was co-transfected with either mCh or [mCh]KIF5 WT in (CV-1-derived) COS-7 cells, precipitating [FLAG]B14 pulled down [mCh]KIF5 WT but not the mCh control protein (Fig. 2b, first and second panels). COS-7 cells were used instead of CV-1 cells due to their higher transfection efficiency. As expected, endogenous Hsc70, Hsp105 and SGTA also co-precipitated with [FLAG]B14 (Fig. 2c). However, other cytosolic HSPs including HspBP1, Hsp27 and HspB5 did not (Fig. 2c). The level of [mCh]KIF5 WT that co-precipitated with [FLAG]B14 was unaffected by SV40 infection (Fig. 2d). These data indicate that the KIF5–B14 interaction occurs in both human and monkey cells, and is specific.

To clarify the nature of the KIF5–B14 interaction, we used a previously established J-domain mutant of B14 (H136Q), which inefficiently recruits the cytosolic Hsc70-SGTA-Hsp105 complex[11,21]. We found that although [FLAG]B14 H136Q associates weakly with the cytosolic chaperone complex when compared with [FLAG]B14 WT, binding between [FLAG]B14 H136Q and [mCh]KIF5 WT remained intact (Fig. 2e). These results suggest kinesin-1 is recruited to B14 in a J-domain-independent manner, although they do not rule out the involvement of other unknown B14 partners in mediating this interaction. In addition to full-length WT KIF5, our results revealed that the truncated KIF5 DN can bind to endogenous B14 (Fig. 2f), suggesting that KIF5's cargo-binding tail domain is likely to be responsible for interacting with B14.

**Kinesin-1 facilitates SV40 translocation into the cytosol.** As kinesin-1 associates with the ER membrane via binding to B14, we asked whether this strategically positions the motor to promote cytosol entry of SV40 from the ER. To monitor cytosol arrival of SV40, we used a previously established cell-based, semi-permeabilized ER-to-cytosol transport assay[17,18]. In this assay, COS-7 cells expressing mCh, [mCh]KIF5 WT or [mCh]KIF5 DN were infected with SV40, harvested and treated with a low concentration of digitonin to permeabilize the plasma membrane without damaging the ER membrane. Cells were then centrifuged to generate two fractions, a supernatant fraction that contains cytosolic proteins and virus that reaches the cytosol ('cytosol' fraction), and a pellet fraction that harbours membranes including the ER as well as virus associated with membranes ('membrane' fraction). The integrity of the fractionation procedure can be monitored in the release of Hsp90 to the cytosolic fraction, and the pelleting of ER-resident protein disulfide isomerase with the membrane fraction (Fig. 3a). Using this assay, we found that expression of [mCh]KIF5 DN but not [mCh]KIF5 WT potently reduced the SV40 VP1 level in the cytosol when compared with cells expressing mCh (Fig. 3a, top panel; the VP1 band intensity is quantified in Fig. 3c). These findings demonstrate that kinesin-1 activity is critical for the transfer of SV40 from the ER to the cytosol.

To evaluate whether [mCh]KIF5 DN expression impairs arrival of the virus to the ER from the plasma membrane, we used a previously established Triton X-100 extraction protocol[17,19] to isolate SV40 that reached the ER from the cell surface (ER-localized fraction). We found that expression of [mCh]KIF5 DN had no significant effect on the level of ER-localized SV40 (Fig. 3b, bottom panel; the VP1 band intensity is quantified in Fig. 3c). Therefore, the impairment in cytosol arrival of SV40 caused by [mCh]KIF5 DN expression (Fig. 3a,c) is not due to a block in virus trafficking from the plasma membrane to the ER. These data further support the idea that the kinesin-1 motor promotes cytosol entry of SV40 from the ER.

Another toxic agent that traffics to the ER and penetrates the ER membrane to reach the cytosol during infection is cholera toxin[30,31]. However, in this case, expressing [mCh]KIF5 DN did not affect cytosol arrival of the toxin (Fig. 3d, top panel). Hence, expressing this DN acting motor tail domain did not globally impair all ER membrane transport processes, indicating that kinesin-1 specifically acts on SV40's ER membrane penetration event.

**Kinesin-1 powers SV40-induced foci formation.** How does kinesin-1 drive SV40 entry into the cytosol from the ER? Previous studies have demonstrated that during SV40 ER membrane penetration, specific ER membrane proteins including BAP31, BAP29, B14, B12 and C18 are reorganized into discrete puncta called foci that serve as cytosol entry sites for SV40 (refs 18–20). As kinesin-1 is associated with B14, we asked whether this motor might power formation of these virus-induced structures. Accordingly, CV-1 cells expressing mCh, [mCh]KIF5 WT or [mCh]KIF5 DN were infected with SV40 (MOI 20) and subsequently immunostained for VP1 and BAP31, an ER membrane protein that accumulates in the virus-induced foci[18]. Strikingly, large virus (VP1)-containing BAP31 foci were detected in cells expressing mCh or [mCh]KIF5 WT, whereas foci in [mCh]KIF5 DN-expressing cells appeared more dispersed and smaller (Fig. 4a). We note that the mCh signal in cells expressing [mCh]KIF5 DN is striated, a commonly observed phenotype that is due to ATP-independent binding of the KIF5 tail domain to microtubule tracks[32–36]. Quantification of focus formation demonstrates that the percentage of cells displaying small foci increased significantly in [mCh]KIF5 DN-expressing cells when compared with cells expressing mCh or [mCh]KIF5 WT (Fig. 4b, MOI 20, compare dark grey bars); concomitantly, the percentage

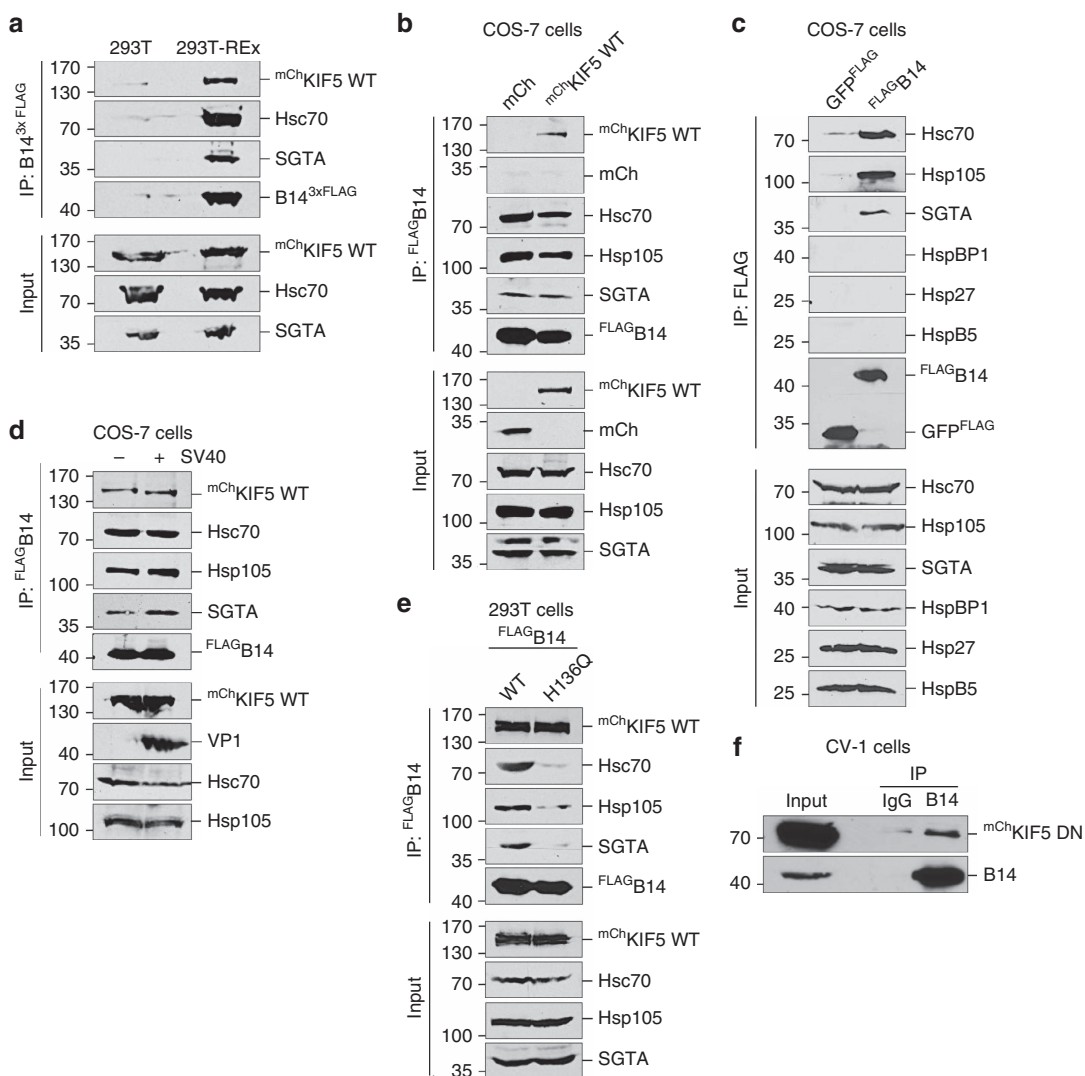

**Figure 2 | Kinesin-1 facilitates SV40 translocation into the cytosol.** (**a**) Flp-In 293T-REx cells expressing B14[3xFLAG] were treated with protein cross-linker dithiobis(succinimidyl proprionate) (DSP), lysed and B14[3xFLAG] was immunoprecipitated. HEK 293T cells not expressing B14[3xFLAG] were used as a control. Bound proteins were eluted and separated by SDS–PAGE followed by immunoblotting with the indicated antibodies. (**b**) COS-7 cells expressing [FLAG]B14 were transiently transfected with mCh or [mCh]KIF5 WT. Cells were treated with DSP, lysed and [FLAG]B14 was immunoprecipitated. Bound proteins were analysed as in **a**. (**c**) COS-7 cells transiently transfected with GFP[FLAG] or [FLAG]B14 were processed as in **b**. The bound proteins were analysed with the indicated antibodies. (**d**) As in **b**, except cells were infected with SV40 MOI 50 (lane marked ' + '). (**e**) 293T cells expressing [FLAG]B14 WT or mutant [FLAG]B14 H136Q were transiently transfected with [mCh]KIF5 WT and immunoprecipitation was performed as in **b**. (**f**) CV-1 cells were transiently transfected with [mCh]KIF5 DN. Cells were lysed and endogenous B14 was immunoprecipitated. Bound proteins were analysed with the indicated antibodies.

of cells with large foci decreased in [mCh]KIF5 DN-expressing cells when compared with mCh- or [mCh]KIF5 WT-expressing cells (Fig. 4b, MOI 20, compare light grey bars). The same trend was found when the experiment was repeated at a lower SV40 concentration (Fig. 4b, MOI 5). We note that the lowest MOI (0.5) for the foci experiments was not used, because foci formation is difficult to observe at this virus concentration. Hereon, we used an MOI 20 for all foci formation assays.

When cells were immunostained for VP2/3 (instead of VP1) proteins, a similar result was observed; these internal viral proteins are exposed only when SV40 reaches the ER and undergoes conformational change[10,12]. Specifically, expressing [mCh]KIF5 DN decreased the large (and increased the small) VP2/3-containing foci when compared with cells expressing mCh or [mCh]KIF5 WT (Fig. 4c; quantified in Fig. 4d). Appearance of the VP2/VP3 signal even in cells expressing [mCh]KIF5 DN demonstrates that inhibition of kinesin-1 function does not

impede SV40 trafficking to the ER from the cell surface, consistent with our previous biochemical analysis (Fig. 3b). To further validate the importance of KIF5 in SV40-induced foci formation, we analysed BAP31 foci formation in KIF5B knockdown CV-1 cells. Our results demonstrate that KIF5B depletion reduced large (and increased small) foci formation (Fig. 4e, quantified in Fig. 4f), consistent with findings using the DN overexpression approach. Thus, these data strongly suggest that the kinesin-1 motor powers foci formation to promote cytosol entry of the virus leading to infection.

Similar to the importance of microtubules at a post-ER arrival step during SV40 infection (Fig. 1), an intact microtubule network is also essential for large focus formation because addition of nocodazole, but not STLC and to a much lower extent MG132, at 5 hpi decreased formation of this structure (Fig. 4g). These findings suggest that generation of virus-triggered large focal structures requires intact microtubule tracks, strengthening

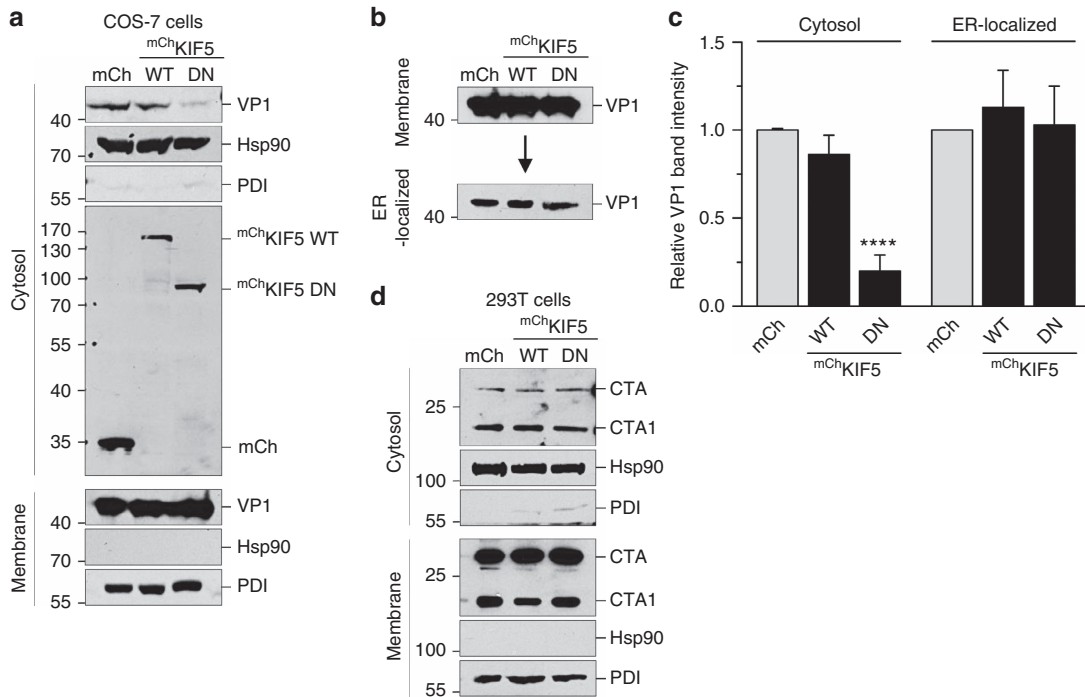

**Figure 3 | Kinesin-1 facilitates SV40 translocation across the ER membrane into the cytosol.** (**a**) COS-7 cells expressing the indicated mCh-tagged constructs were infected with SV40 (MOI 5). Cells (12 hpi) were harvested and processed using the semi-permeabilized cytosol arrival assay (see Methods). Hsp90 and protein disulfide isomerase (PDI) serve as markers for the cytosol and membrane fractions, respectively. (**b**) Membrane fraction in **a** was solubilized in a buffer containing 1% Triton X-100. After centrifugation, the extracted material containing ER-localized SV40 was analysed by immunoblotting with VP1 antibodies. (**c**) Relative VP1 band intensities in the cytosol (**a**) and ER-localized (**b**) fractions were quantified. Data are normalized to mCh (control). Values represent average of the mean ($n = 5$) ± s.d. $P$-values were calculated with the two-tailed Student's $t$-test; ****$P < 0.0001$. (**d**) As in **a**, except 293T cells were treated with 10 nM cholera toxin (instead of SV40) for 90 min before being processed and analysed as in **a**.

the idea that the microtubule-based motor kinesin-1 is required to generate these membrane penetration sites essential for SV40 cytosol entry.

**Restoring kinesin-1 rescues foci maturation and infection.** Our data led us to hypothesize that the small foci are immature structures requiring kinesin-1 activity to mature into the large foci. If expression of the DN kinesin-1 simply precludes small-to-large foci maturation, we reasoned that restoring kinesin-1 activity by connecting the motor-less kinesin-1 ([mCh]KIF5 DN) to its corresponding motor domain (KIF5 motor) should coalesce small foci into large foci and, as a consequence, rescue infection. To test this, we used a previously established FK506-binding protein (FKBP)–FKBP rapamycin binding (FRB) chemical-induced dimerization strategy[37]. In this approach, the FKBP domain is appended to [mCh]KIF5 DN forming [FKBP-mCh]KIF5 DN, whereas FRB is attached to the KIF5C motor domain to generate KIF5 motor[FRB]. Addition of the rapamycin analogue-1 (rapa) (induces FKBP–FRB dimerization, thereby generating full-length active KIF5 (ref. 37). A diagram of this 'split' kinesin-1 strategy is shown in Fig. 5a. The expression levels of mCh, [FKBP-mCh]KIF5 DN, KIF5 motor[FRB] or [FKBP-mCh]KIF5 DN and KIF5 motor[FRB] in CV-1 cells are shown in Fig. 5b. To evaluate the FKBP–FRB dimerization approach in our system, we found that only in the presence of the rapa, immunoprecipitation of [FKBP-mCh]KIF5 DN significantly pull down KIF5 motor[FRB] (Supplementary Fig. 2). These findings validate the integrity of the FKBP-FRB dimerization system.

Using the experimental set-up depicted in Fig. 5c, we first examined SV40 infection in cells expressing [FKBP-mCh]KIF5 DN with or without the KIF5 motor[FRB] in the presence or absence of the rapa linker. Consistent with expressing [mCh]KIF5 DN,

expression of [FKBP-mCh]KIF5 DN also blocked SV40 infection (Fig. 5d). However, in [FKBP-mCh]KIF5 DN-expressing cells, co-expression of KIF5 motor[FRB] fully rescued infection only if rapa was added—this occurred regardless of whether the linker was incubated at infection (0 hpi) or 5 hpi (Fig. 5d). These results demonstrate that restoring kinesin-1 activity at a post-ER arrival step is sufficient to rescue SV40 infection.

We next assessed whether the ability to rescue infection by restoring kinesin-1 activity correlates with maturation of the small foci into the large foci. In cells expressing [FKBP-mCh]KIF5 DN, a higher percentage of the cells harbored small foci (Fig. 5e, top row) when compared with cells expressing mCh (quantified in Fig. 5f), similar to the effect of expressing [mCh]KIF5 DN (Fig. 4a,b). Importantly, co-expressing KIF5 motor[BFP-FRB] (where blue fluorescent protein (BFP) is spliced in between the motor and FRB) in these cells along with adding the rapa linker (at infection or 5 hpi) almost completely restored large foci formation (Fig. 5e, bottom row; quantified in Fig. 5f). These findings demonstrate that restoring kinesin-1 activity promotes foci maturation, mirroring the effect on SV40 infection (Fig. 5d).

Maturation of foci upon restoring kinesin-1 activity was also observed using live-cell microscopy according to the experimental set-up depicted in Fig. 5g. We note that in addition to expressing [FKBP-mCh]KIF5 DN and KIF5 motor[FRB], fluorescently tagged BAP31 ([GFP]BAP31) was also co-expressed to observe foci maturation in real time. Under the experimental condition (cells expressing [GFP]BAP31, [FKBP-mCh]KIF5 DN and KIF5 motor[FRB]), we observed a seven fold increase in fusion events, leading to formation of the single large BAP31 focus when compared with the control condition (cells expressing [GFP]BAP31 and [FKBP-mCh]KIF5 DN) (Fig. 5h; see also Supplementary Movie 1). These

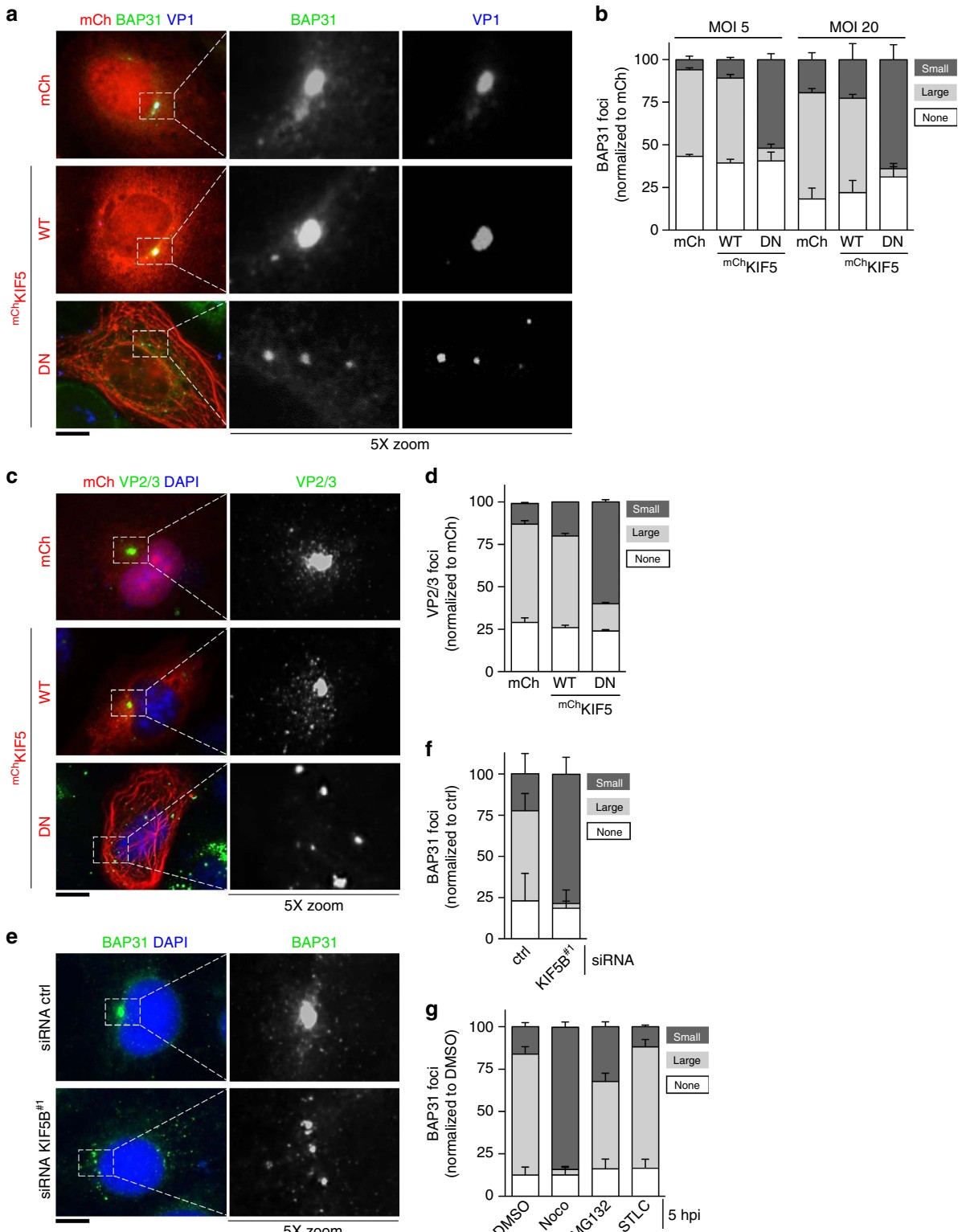

**Figure 4 | Kinesin-1 powers SV40-induced foci formation.** (**a**) CV-1 cells expressing the indicated mCh constructs were infected with SV40 (MOI 20). Cells (16 hpi) were fixed, stained for BAP31 and VP1, and analysed by immunofluorescence microscopy. The middle and right panels show 5X-enlarged images of the boxed regions. Scale bar, 10 μm. (**b**) CV-1 cells expressing the indicated mCh constructs were infected with SV40 (MOI 5 or 20). Cells were processed as in **a** and quantified for the presence of large (dark grey bars) or small (light grey bars) BAP31-positive foci. Values represent average of the mean ($n = 3$) ± s.d. (**c**) As in **a**, except cells were immunostained for VP2/3. Scale bar, 10 μm. (**d**) As in **b**, except cells were quantified for VP2/3-positive foci. Values represent average of the mean ($n = 3$) ± s.d. (**e**) Control (ctrl) or KIF5B[#1] siRNA-transfected CV-1 cells were infected with SV40 (MOI 20) for 16 h. Samples were processed and analysed as in **a**. Scale bar, 10 μm. (**f**) Quantification of the BAP31-positive foci in **e**. Values represent average of the mean ($n = 4$) ± s.d. (**g**) CV-1 cells were infected with SV40 (MOI 20) after 5 hpi, cells were treated with different drugs (1 μM nocodazole, 20 μM MG132 or 1 μM STLC). Cells (16 hpi) were fixed and immunostained against BAP31. The sizes of the BAP31-positive foci in each cell were scored as either large (dark grey bars) or small (light grey bars). Values represent average of the mean ($n = 3$) ± s.d.

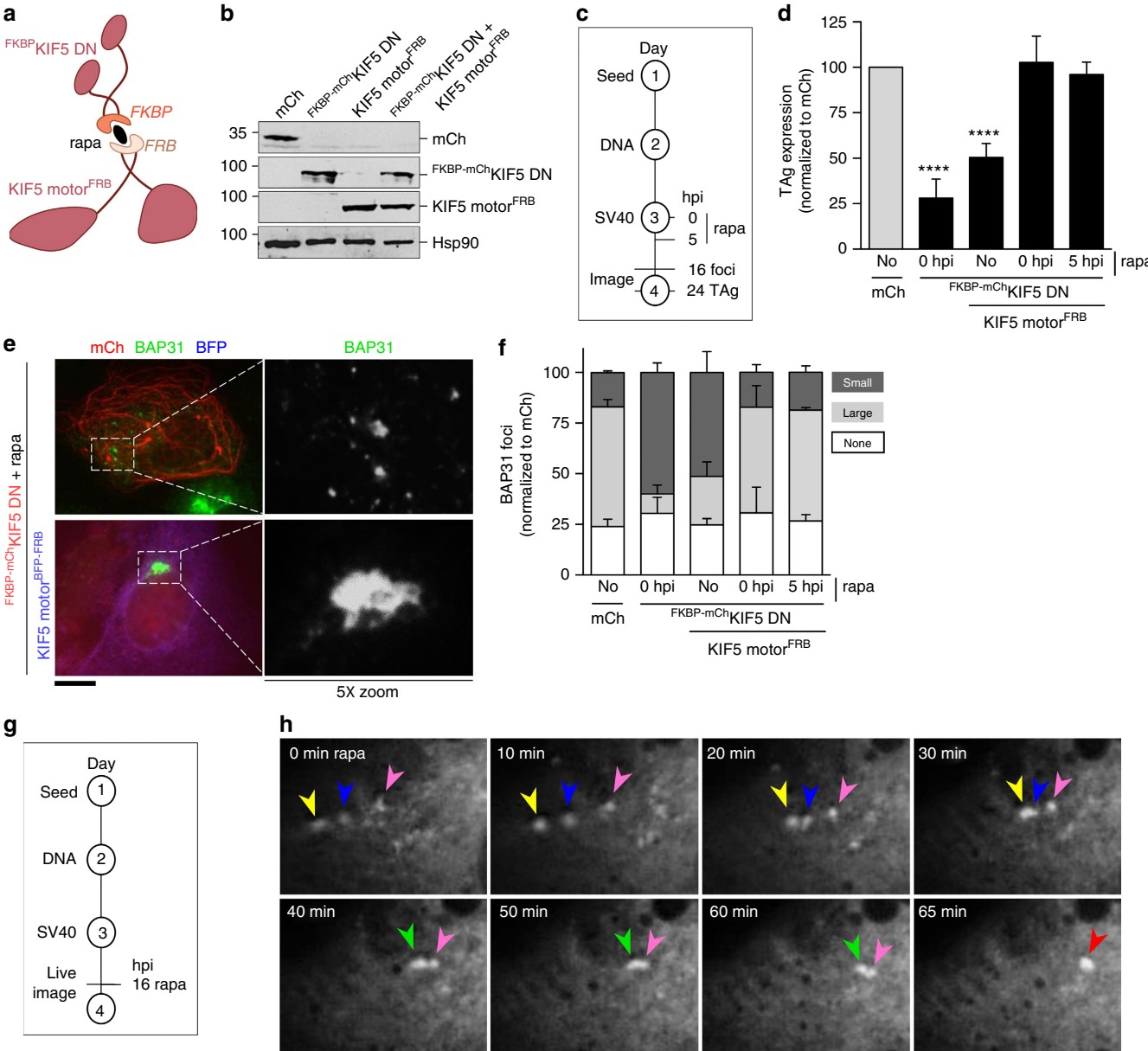

**Figure 5 | Restoring kinesin-1 rescues foci maturation and infection.** (**a**) A diagram depicting the split kinesin-1 strategy. (**b**) CV-1 cells expressing the indicated constructs were lysed and immunoblotted with the indicated antibodies. (**c**) Depiction of the experimental set-up used in **d**–**f**. (**d**) CV-1 cells expressing the indicated constructs were infected with SV40 (MOI 0.5). Cells (0 or 5 hpi) were treated with the rapa linker. Cells (24 hpi) were fixed and immunostained against TAg. Cells expressing mCh-tagged proteins were scored for TAg positive signal by immunofluorescence microscopy. Data are normalized to the mCh control (grey bar). Values represent average of the mean ($n \geq 3$) ± s.d. *P*-values were calculated with the two-tailed Student's *t*-test; ****$P < 0.0001$. (**e**) Representative images of SV40-infected cells (MOI 20) expressing the indicated proteins. The right panels show 5X-enlarged images of the boxed regions. Scale bar, 10 μm. (**f**) Cells from **e** were quantified for the presence of large (dark grey bars) or small (light grey bars) BAP31-positive foci. Values represent average of the mean ($n \geq 3$) ± s.d. (**g**) Depiction of the live cell imaging experiment in **h**. (**h**) Live-cell imaging of BAP31-positive foci. Time sequences of the images are shown at the top left corner and indicate time after addition of rapa. The different colour arrows (yellow, blue and pink) represent small foci that fuse into a larger focus (yellow and blue to green and then green and pink to red) over time. Scale bar, 10 μm.

results further support the notion that functional kinesin-1 is exploited to power SV40-induced foci maturation.

**Kinesin-2 and -3 do not promote foci maturation or infection.** We asked whether recruiting the motor domains of other kinesin family members to motor-less kinesin-1 also promotes SV40 foci maturation and infection. To this end, we co-expressed the motor domains of kinesin-1 (KIF5 motor[BFP-FRB]), kinesin-2 (KIF17 motor[BFP-FRB])[38] or kinesin-3 (KIF1A motor[BFP-FRB])[39]

with [FKBP-mCh]KIF5 DN and added rapa at 0 hpi. This setup connects the motor domain of kinesin-1, kinesin-2 or kinesin-3 to the kinesin-1 tail domain, as depicted in Fig. 6a. Importantly, previous reports established that the induced dimerization system faithfully recapitulates motor-dependent transport processes[40,41]. We validated the integrity of the FKBP–FRB dimerization system for the KIF17 and KIF1A motors by co-immunoprecipitation experiments (Supplementary Fig. 2). Using this system, we found that in cells expressing [FKBP-mCh]KIF5 DN, only co-expression of

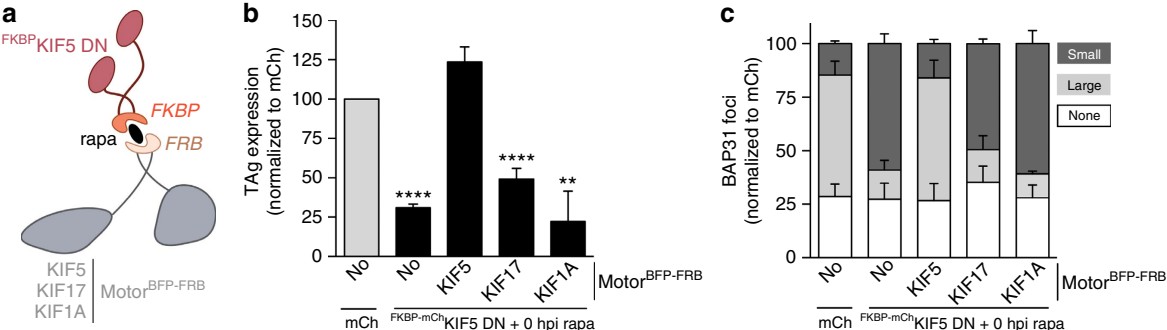

**Figure 6 | Kinesin-2 and kinesin-3 do not promote SV40 foci maturation and infection.** (**a**) A diagram depicting the experimental set-up used in **b**,**c**. (**b**) CV-1 cells expressing the indicated constructs were infected with SV40 (MOI 0.5). Cells were treated with the rapa linker at infection (0 hpi rapa). Cells (24 hpi) were fixed and immunostained against TAg. Cells expressing mCh-tagged proteins were scored for TAg-positive signal by immunofluorescence microscopy. Data are normalized to the mCh control (grey bar). Values represent average of the mean ($n = 3$) ± s.d. $P$-values were calculated with the two-tailed Student's $t$-test; **$P < 0.01$ and ****$P < 0.0001$. (**c**) As in **b**, except cells were infected with SV40 (MOI 20) for 16 h and quantified for the presence of large (dark grey bars) or small (light grey bars) BAP31-positive foci. Values represent average of the mean ($n = 3$) ± s.d.

KIF5 motor[BFP-FRB] but not KIF17 motor[BFP-FRB] or KIF1A motor[BFP-FRB] in the presence of the rapa linker rescued SV40 infection (Fig. 6b) and foci maturation (Fig. 6c). These results indicate that the motor domain of kinesin-1 is specifically required to promote SV40-induced foci maturation leading to cytosol arrival and infection.

**Kinesin-1 selects acetylated microtubules for foci formation.** We envisioned two possibilities to account for this specific requirement for kinesin-1 motor activity. First, we hypothesized that kinesin-1 is uniquely equipped to drive foci formation, because it is a 'stubborn' motor and walks processively under considerable hindering loads; in contrast, kinesin-2 and kinesin-3 motors readily detach from the microtubule track under load[42–46]. To test this, we generated a series of 'weakened' kinesin-1 motor domains by deleting 3, 4 or 5 amino acids (Δ3–Δ5) from the cover strand, an element of the kinesin-1 motor domain that is essential for force generation[47–49]. Surprisingly, these weakened motor domains were able to rescue SV40 infection to the same extent as the WT KIF5 motor (Fig. 7b). Thus, the requirement for kinesin-1 during SV40 infection is not due to the unique ability of this motor to sustain transport under high load conditions.

Second, we hypothesized that kinesin-1 is uniquely equipped to drive foci formation because of its ability to select specific populations of microtubule tracks. Previous studies demonstrated that kinesin-1 motors respond to the tubulin code and move preferentially on posttranslationally modified microtubules in cells, including acetylated and detyrosinated microtubules, whereas kinesin-2 and kinesin-3 motors are not selective[39]. To test whether virus-induced foci are associated with acetylated microtubules, infected CV-1 cells were immunostained for both BAP31 and acetylated α-tubulin. We found that the majority of BAP31-containing foci localize to the central region of the microtubule network that is dense with acetylated α-tubulin (Fig. 7c, bottom row). Importantly, the large SV40-induced BAP31 focus does not co-localize with MTOC (γ-tubulin), Golgi (Giantin) or the early endosomes (EEA1) (Supplementary Fig. 3). To test whether kinesin-1's ability to undergo preferential transport along modified microtubule tracks facilitates foci formation and SV40 infection, we treated cells with taxol, a drug that increases microtubule modifications including α-tubulin acetylation[50,51] (Fig. 7d). Addition of taxol for 20 min after the virus has reached the ER (at 5 hpi, Fig. 7e) caused a significant increase in virus infection (Fig. 7f,g). Consistent with this, the extent of foci formation also increased in taxol-treated cells (Fig. 7h).

To confirm that the increased infection upon taxol treatment was specific to kinesin-1's ability to transport along the modified microtubule tracks, we compared the ability of kinesin-1, kinesin-2 and kinesin-3 motors to drive SV40 infection under the taxol-treated condition (Supplementary Fig. 4a). We found that expressing [FKBP-mCh]KIF5 DN potently blocked SV40 infection in the presence of taxol. Importantly, addition of the rapa to [FKBP-mCh]KIF5 DN-expressing cells can rescue SV40 infection only in cells co-expressing the KIF5 motor[FRB] but not the KIF17 motor[FRB] or KIF1A motor[FRB] domains (Supplementary Fig. 4b). These findings are consistent with our observations in Figs 6 and 7, strengthening the idea that SV40 specifically co-opts kinesin-1's unique ability to select modified microtubule tracks to drive foci maturation essential for infection.

As taxol treatment can lead to different types of microtubule modifications[50,52], we further evaluated the importance of acetylated microtubules in SV40 infection using tubacin, a membrane-permeable inhibitor of the microtubule deacetylase HDAC6 (ref. 53) (Supplementary Fig. 4c). Treatment of CV-1 cells with tubacin increased α-tubulin acetylation (Supplementary Fig. 4d), as anticipated. Under this condition, we observed an increase in TAg expression by both immunoblotting and immunofluorescence analyses (Supplementary Fig. 4e,f), as well as an enhancement of BAP31 foci formation (Supplementary Fig. 4g). These results are consistent with the taxol data, supporting the basic premise that the preferential motility of kinesin-1 motors along acetylated microtubules promotes foci formation required for successful SV40 infection.

**Discussion**
Successful viral entry requires penetration of a host membrane, a process that remains mysterious for non-enveloped viruses. For SV40, a non-enveloped virus, the ER membrane must be breached to gain access to the cytosol and cause infection[10–12,18–21,54]. A major gap in our understanding is whether the virus hijacks a preformed channel or remodels the ER membrane to create a membrane penetration site, to access the cytosol. A series of reports support the latter scenario in which SV40 was found to remodel the ER membrane by inducing reorganization of several ER transmembrane proteins into discrete puncta called foci[18–21]— these structures in turn serve as sites from where the virus enters the cytosol. Although this provides the first example of a viral particle creating its own membrane penetration site, a key question is how this sub-organellar structure is constructed. Our findings here reveal two new aspects of how SV40 co-opts the cellular machinery to

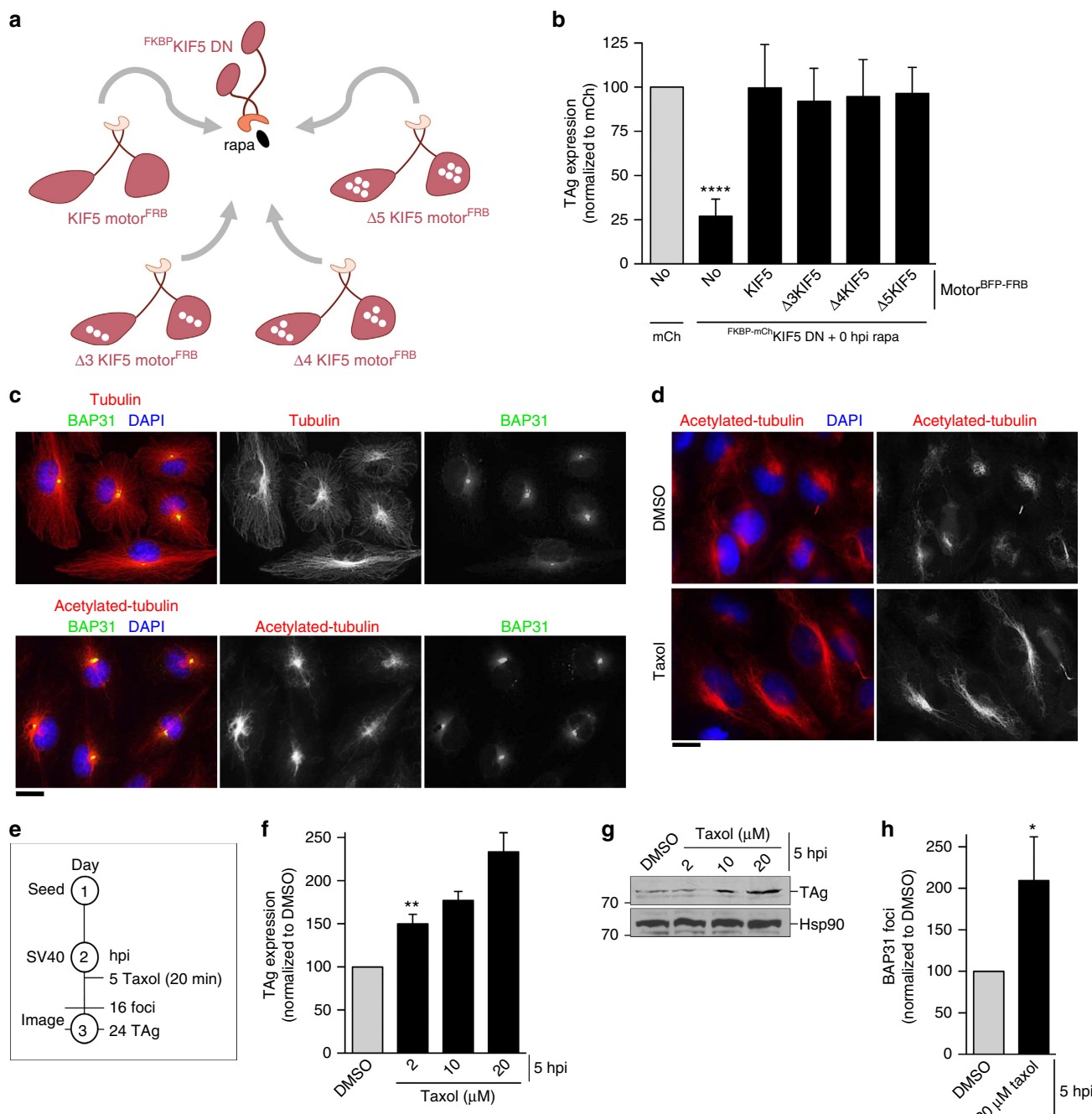

**Figure 7 | Kinesin-1 selects acetylated microtubules for foci formation.** (**a**) A diagram depicting the experimental set-up used in **b**. White empty circles represent the N-terminal deletion of 3, 4 or 5 amino acids from the KIF5 motor domain. (**b**) CV-1 cells expressing the indicated constructs were infected with SV40 (MOI 0.5). Cells were treated with the rapa linker at infection (0 hpi rapa). Cells (24 hpi) were fixed and immunostained against TAg. Cells expressing mCh-tagged proteins were scored for TAg-positive signal by immunofluorescence microscopy. Data are normalized to the mCh control (grey bar). Values represent average of the mean ($n \geq 3$) ± s.d. P-values were calculated with the two-tailed Student's t-test; ****P < 0.0001. (**c**) SV40-infected CV-1 cells were immunostained with the indicated antibodies. Scale bar, 20 μm. (**d**) Representative images of CV-1 cells treated with 10 μM taxol or DMSO for 20 min. Scale bar, 20 μm. (**e**) Depiction of the experimental setup used in **f**,**g**. (**f**) Cells were infected with SV40 (MOI 0.5). Cells (5 hpi) were treated with varying concentrations of taxol or DMSO for 20 min and washed. Cells (24 hpi) were fixed and immunostained against TAg. TAg-positive cells were counted by immunofluorescence microscopy and normalized to DMSO control (grey bar). Values represent average of the mean ($n \geq 3$) ± s.d. P-values were calculated with the two-tailed Student's t-test; **P < 0.01. (**g**) As in **f**, except SV40-infected cells were treated with 20 μM of taxol and 48 hpi, cells were lysed and the resulting whole-cell lysate was immunoblotted with the indicated antibodies. (**h**) As in **f**, except 16 hpi, cells were quantified for the presence of BAP31-positive foci. Values represent average of the mean ($n \geq 3$) ± s.d. P-values were calculated with the two-tailed Student's t-test; *P < 0.05.

penetrate the ER membrane. First, we demonstrate that microtubule-based motility is required for the movement and subsequent coalescence of ER-localized viral particles into large focal structures essential for membrane penetration. Second, we demonstrate that the kinesin-1 molecular motor is specifically selected for this transport as its unique property of selecting modified microtubule tracks enables the spatial restriction of the viral particle to the perinucleus, as depicted in Fig. 8.

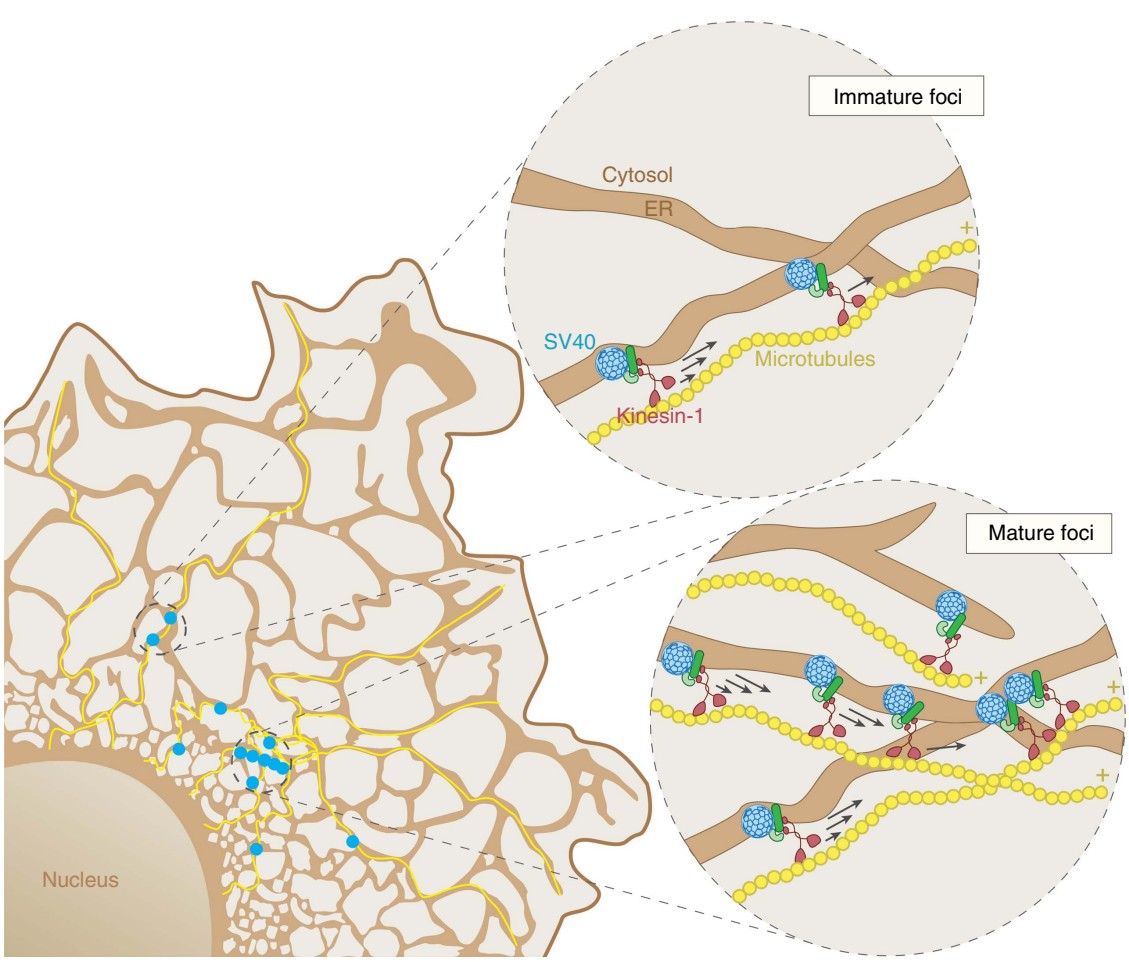

**Figure 8 | SV40 exploits kinesin-1's activity to promote foci maturation.** During SV40 ER membrane penetration, the virus (blue) engages the ER membrane J-protein B14 (dark green), which in turn recruits the cytosolic Hsc70-Hsp105-SGTA extraction complex (light green) and kinesin-1 (purple). This motor then powers the local movement of SV40, B14, and the cytosolic chaperones present in 'small immature foci' (magnified circle on the top), reorganizing them into 'large mature foci' (magnified circle on the bottom). These large focal structures are predominantly found proximal to the nucleus where acetylated microtubules (bent and curved yellow lines) are enriched. Formation of mature foci locally concentrates the cytosolic chaperones to enable efficient extraction of the viral particle into the cytosol.

Our biochemical analyses unveiled a physical connection between the ER membrane J-protein B14 and the cytosolic kinesin-1 motor. Although the precise nature of the B14–kinesin-1 interaction remains unclear, this interaction occurs independent of B14's J-domain, a motif used to recruit the cytosolic Hsc70-SGTA-Hsp105 complex that extracts SV40 into the cytosol[11,21]. It is possible that unidentified cellular adapters facilitate complex formation between these two proteins. Our analyses further revealed that kinesin-1 is recruited to B14 in the absence of virus infection, suggesting that binding between kinesin-1 and B14 subserves an endogenous function. Kinesin-1 has previously been implicated in ER tubule extension along microtubules[28], perhaps via association with the ER membrane proteins kinectin and/or protrudin[55–58]. The interaction with B14 (and its binding partner B12) thus implies an additional kinesin-1 function in major ER quality control pathways including ERAD and ER-associated autophagy[59–61].

By expressing an established KIF5 DN (motor-less kinesin-1) and using a knockdown strategy in the context of cell-based assays, our results demonstrate that kinesin-1 promotes SV40 ER membrane-to-cytosol transport leading to successful infection, without affecting viral trafficking from the plasma membrane to the ER. By contrast, this motor does not appear to be involved in murine PyV infection[62]. Although this virus shares a similar entry pathway as SV40 (ref. 4), it is not known whether murine PyV co-

opts B14 or creates foci to access the cytosol. The key finding that kinesin-1 executes a critical role during SV40 infection is not without precedent—this motor is also used by other viruses including the human adenovirus-2/5, human immunodeficiency virus-1 and herpes simplex virus-1 during entry[22–24].

To dissect the molecular basis by which kinesin-1 promotes SV40 cytosol entry from the ER, we used a chemical-induced dimerization strategy and found that this motor is essential for maturation of the foci. In this process, kinesin-1's cargos consist of structural components of the foci including B14, B12, C18 and BAP31. Motor-driven, microtubule-based transport of these cargos enables them to come in contact and accumulate, gradually assembling into a mature structure. Precisely when kinesin-1's activity is needed to mobilize these cargos during foci maturation is unclear. However, as kinesin-1 is responsible for transforming small to large foci, we posit that formation of the small foci is unlikely to require kinesin-1's activity (Fig. 8). Another outstanding question is the nature of the virus-induced signal that activates kinesin-1's motor activity to initiate cargo transport. Can the virus directly bind to and enhance the motor activity, or does it trigger a signalling cascade that in turn stimulates kinesin-1's function? Future experiments are required to address these questions. Interestingly, in contrast to promoting an assembly event during SV40 infection, kinesin-1 can support a disassembly reaction in the case of adenovirus entry[24]; in this

instance, kinesin-1 is able to terminally disassemble the viral particle to liberate the viral genome for nuclear import and subsequent replication.

Our analyses further revealed that other kinesin motors cannot replace kinesin-1's function in SV40-induced foci formation, as only the motor domain of kinesin-1, but not kinesin-2 or kinesin-3, was able to rescue foci formation and infection. The selection of kinesin-1 to build the foci does not appear to require this motor's unique property of persistent transport under load. Instead, the kinesin-1 motor domain's unique ability to selectively recognize modified microtubules provides at least part of the explanation. Kinesin-1's motor domain is known to preferentially transport along acetylated and detyrosinated microtubules in cells[39,63–65], modifications that are predominantly found on microtubules in a cell's perinuclear region[50,66] where SV40-induced foci are typically found. In fact, increasing the levels of acetylated modified microtubules stimulated virus-induced foci formation and infection, strengthening the notion that kinesin-1's selective recognition of modified microtubules is important for its ability to promote foci maturation essential for stimulating infection. These findings support the idea of a 'tubulin code' wherein post-translational modifications of tubulin subunits within specific microtubule populations provide spatial cues for microtubule-based activities[67,68].

As a plus end-directed microtubule motor, kinesin-1 generally transports cargos from a cell's centre to its periphery[69]. SV40's transport from the ER to the cytosol (and then the nucleus) might therefore appear to move opposite to the direction guided by kinesin-1. To reconcile this, we envision that within the convoluted ER membranous network, local transport of foci components powered by kinesin-1 can occur in a plus-end direction along modified sections of microtubules that do not extend to the cell periphery. In agreement with this idea, acetylated microtubule tracks localize to the perinucleus due to their propensity to bend and curve[70], and the use of these modified microtubules by kinesin-1 to form foci should deposit the virus in the cytosol proximal to the nucleus, enabling efficient nuclear entry leading to infection. Such a scenario elegantly illuminates how a virus exploits a unique relationship between a specific molecular motor and its track to catalyse the cytosol entry process.

## Methods

**Reagents.** CV-1 (catalogue: CCL-70), COS-7 (catalogue: CRL-1651) and HEK 293T (catalogue: CRL-3216) cells were purchased from ATCC. Cells were grown in complete DMEM medium (cDMEM; 10% fetal bovine serum, 10 U ml$^{-1}$ penicillin and 10 µg ml$^{-1}$ streptomycin; Gibco, Grand Island, NY). Opti-MEM and 0.25% trypsin-EDTA were purchased from Gibco. Flp-In 293 TRex cells (catalogue: R78007, Invitrogen, Carlsbad, CA) transfected with pcDNA-B14$^{3xFLAG}$ were selected in cDMEM containing blasticidin and hygromycin (Invitrogen). B14$^{3xFLAG}$ was expressed to near endogenous level by overnight induction with freshly prepared 5 ng ml$^{-1}$ tetracycline (Sigma, St Louis, MO).

**Chemicals and antibodies.** Nocodazole, STLC, taxol (Paclitaxel), Triton X-100, phenylmethanesulfonylfluoride (PMSF), N-ethymaleimide and anti-FLAG M2 antibody conjugated agarose beads were purchased from Sigma; protein A/G-conjugated agarose beads, dithiobis(succinimidyl proprionate) and MG132 from ThermoFisher (Rockford, IL), tubacin from Enzo Lifesciences (Farmingdale, NY), rapalog-1 or A/C heterodimerizer from Clonetech (catalogue: 635057; Mountain View, CA) and digitonin from EMD Millipore (San Diego, CA). Antibodies used in this study are listed in Supplementary Table 1.

**Preparation of SV40.** SV40 was purified as described previously[17]. Briefly, CV-1 cells were transfected with pUCSV40 encoding SV40 genome (Gene bank: J02400.1) (gift from Dr H. Handa, Tokyo Medical University). Cells were harvested and lysed in a buffer containing 50 mM Hepes pH 7.5, 150 mM NaCl and 0.5% Brij 58 for 30 min on ice, and the supernatant was collected after centrifugation at 20,000 g for 10 min. A discontinuous 20% and 40% OptiPrep gradient (60% stock solution of iodixanol in water; Sigma) was prepared and the supernatant was placed on top of the gradient. Tubes are centrifuged at 49,500 r.p.m. for 2 h at 4 °C in an

SW55Ti rotor (Beckman Coulter, Indianapolis, IN). A white interface formed between 20% and 40% OptiPrep was collected and aliquots were stored at − 80 °C for future use.

**Plasmids.** The source of plasmids $^{FLAG}$B14 WT, $^{FLAG}$B14 H136Q and $^{GFP}$BAP31 are described[19]. Vector pmCherry-C1 (Clonetech) was used as the vector for plasmid $^{mCh}$KIF5 WT and DN. The plasmid for expression of motor-less KIF5 DN was generated by subcloning amino acids 568–964 of rat KIF5B into the mCh-C1 vector. The plasmid $^{FKBP-mCh}$KIF5 DN was then generated by subcloning FKBP-coding sequences N-terminal to the mCh sequence. Plasmids for expression of the following kinesin motor domains have been described[39]: kinesin-1 is rat KIF5C(1–559) (NP_001101200), kinesin-2 is human KIF17(1–488) (NP_065867) and kinesin-3 is rat KIF1A(1–393) (XP_017459420). Motors tagged with BFP and FRB were created by subcloning the relevant protein sequences using PCR or synthetic DNA pieces. For SV40 infection rescue experiments, the empty pCIG vector encoding GFP translated from IRES was used as a control and was a gift from Dr Benjamin Allen (University of Michigan). The rescue plasmid driving the expression of the full-length KIF5C construct (NP_001101200.1) was generated by subcloning the KIF5C open reading frame into the multiple cloning site of pCIG. This plasmid simultaneously expresses both KIF5C and GFP.

**DNA transfection.** For transfection in CV-1 cells, 50% confluent cells in 6 cm plates, 10 or 15 cm dishes were transfected with plasmid using the FuGENE HD (Promega, Madison, WI) transfection reagent at a ratio of 1:4 (plasmid to transfection reagent; w/v). If required, for each construct, the amount of DNA was normalized based on the protein expression by immunoblotting. Cells were allowed to express the protein for at least 24 h before experimentation. For COS-7 and HEK 293T cells in 6 cm plate, polyethylenimine (Polysciences, Warrington, PA) was used as the transfection reagent.

**siRNA transfection.** KIF5 was knocked down using a custom stealth KIF5B-specific siRNA generated and purchased from Invitrogen. The target DNA sequence for siRNA KIF5B$^{#1}$ is 5′-GAG CAC AAG AGA AAG TCC ATG AAA T-3′ and siRNA KIF5B$^{#2}$ is 5′-CAA GCA AGA CAA GAC TTG AAG GGT T-3′. CV-1 cells were reverse transfected twice with 25 nM KIF5B siRNA or a negative control siRNA (ctrl) (Qiagen all-star negative catalogue: 1027281) using Lipofectamine RNAiMAX reagent (Invitrogen) for at least 48 h. The ratio of siRNA to transfection reagent was maintained at 1:4 v/v. For rescue experiments, CV-1 cells were reverse transfected with 25 nM KIF5B$^{#1}$ siRNA with RNAiMAX and after 24 h, cells were again transfected with 25 nM KIF5B$^{#1}$ siRNA and 1.6 µg KIF5C DNA using Lipofectamine 2000 reagent. The ratio of DNA to transfection reagent was maintained at 1:4 v/v.

**Immunoprecipitation.** Cells expressing the transfected proteins were harvested and cell pellets were washed three times with cold PBS (Gibco). Washed cells were incubated with freshly prepared 2 mM dithiobis(succinimidyl proprionate) for 30 min at room temperature with intermittent shaking. This membrane permeable, amine-reactive and thiol-cleavable cross-linker was used to stabilize transient or weak protein–protein interactions. Excess cross-linker was quenched with 200 mM Tris pH 7.5. Cells were then lysed with 1% Triton X-100 in TSEP buffer (50 mM Tris-Cl pH 7.5, 150 mM NaCl, 1 mM EDTA and 1 mM PMSF) at 4 °C for 30 min. Cell lysate was clarified by centrifugation at 16,100 g for 10 min at 4 °C. The resulting supernatant was immunoprecipitated with anti-FLAG-conjugated agarose beads for 2 h at 4 °C. Beads were washed three times with 0.1% Triton X-100 in TSEP buffer. Samples were eluted with 1 × SDS sample buffer with 1.25% β-mercaptoethanol (Sigma) and boiled for 10 min at 95 °C before being subjected to SDS–PAGE and immunoblotting. For immunoprecipitation of endogenous B14, cell lysates derived from CV-1 cells were incubated with either IgG control or B14 antibody overnight at 4 °C. Samples were incubated with protein A/G-conjugated agarose beads for 2 h at 4 °C and processed as above. For immunoprecipitation of FKBP- or FRB-tagged proteins, COS-7 cells expressing the indicated constructs were treated with the rapa linker. After 24 h, cells are lysed with 1% Triton X-100 in TSEP buffer at 4 °C for 30 min. The resulting supernatant was immunoprecipitated using mCh antibody for 2 h at 4 °C, followed by protein A/G beads for 2 h. Beads were washed three times with 0.1% Triton X-100 in TSEP buffer. Samples were eluted and processed as above before being subjected to SDS–PAGE and immunoblotting with a FRB antibody.

**Semi-permeabilized cytosol arrival assay.** This assay is performed as described previously[17,19] with minor modifications. Briefly, COS-7 cells expressing transfected proteins were prechilled at 4 °C for 20 min before infecting with SV40 (MOI 5) for 2 h at 4 °C. Cells were washed and incubated for 12 h at 37 °C. Post infection, cells were lysed in HNP buffer (50 mM Hepes pH 7.5, 150 mM NaCl and 1 mM PMSF) containing 0.025% digitonin and 10 mM N-ethymaleimide at 4 °C for 10 min. The lysate was centrifuged at 16,100 g for 10 min at 4 °C and the supernatant (cytosol) and pellet (membrane) fractions were collected. ER-localized SV40 was isolated by further treatment of the pellet fraction with HNP buffer containing 1% Triton X-100 for 10 min at 4 °C and centrifuged at 20,000 g for

10 min at 4 °C. The fractions were analysed by non-reducing and reducing SDS–PAGE. To assess cytosol arrival of cholera toxin A1 subunit, 293T cells were treated with 10 nM CT (EMD Millipore) for 90 min. Cells were harvested and fractionated as above. For examining SV40 ER arrival in nocodazole-treated CV-1 cells, cells are treated with nocodazole at either 0 hpi or 5 hpi. Cells are processed as above, with the resulting lysate analysed by non-reducing or reducing SDS-PAGE (see figure legend to Supplementary Fig. 1).

**Immunofluorescence microscopy.** CV-1 cells grown on sterile cover slips were infected with SV40 MOI 0.5 (for TAg expression studies) or MOI 20 (for foci formation studies) for 24 h and 16 h, respectively. Infected cells were fixed with 1% formaldehyde for 15 min at room temperature followed by permeabilization with 0.2% Triton X-100 in PBS for 5 min. Cells were then blocked with blocking buffer containing 5% milk in TBST (Tris buffered saline with 0.02% Tween 20) for 15 min. Immunostaining was performed with primary antibody diluted in blocking buffer for 1 h at room temperature and then washed five times with blocking buffer. Cells were incubated with fluorescence dye-conjugated secondary antibody for 30 min at room temperature. Cells were subsequently washed three times with blocking buffer, PBS and water before air drying and mounting on glass slides (Fisher) using ProLong gold (Invitrogen) with or without 4,6-diamidino-2-phenylindole (Molecular Probes, Eugene, OR). Slides were allowed to dry in the dark at room temperature for at least 12 h before imaging. Images were taken using an inverted epifluorescence microscope (Nikon Eclipse TE2000-E, Melville, NY) equipped with × 40, × 60 and × 100 1.40 numerical aperture objectives and standard 4,6-diamidino-2-phenylindole (blue), fluorescein isothiocyanate (green) and TRITC (red) filter cubes. Images were processed using the ImageJ software version 1.48i (NIH).

**SV40 infection and foci formation assays.** For SV40 infection assays, samples were stained for expression of TAg. Infected (TAg-positive) cells were identified by the presence of dense nuclear-localized fluorescence, which is absent in the uninfected cells. For the foci formation assays, cells were stained for endogenous BAP31 and cells harbouring a single BAP31 focus is counted as large foci, while the presence of smaller and dispersed focus is scored as small foci. For drug-treated or siRNA-mediated knockdown experiments, TAg expression and BAP31 foci formation were analysed by scoring at least 250 cells per experiment using the ImageJ programme (Plugin: Cell counter). For DNA transfection experiments, cells expressing fluorescent-tagged proteins were scored under a microscope with an eyepiece. At least 100 cells were scored for each experiment. Data are normalized to the control and represent the average of the mean values from at least three independent experiments ($n \geq 3$) and the error bar represents s.d.

**Taxol- and tubacin-treated experiments.** CV-1 cells were treated with varying concentrations of taxol (0, 2, 10 and 20 μM) for 20 min or 5 μM of tubacin for 4 h. Post-treatment, cells were washed with fresh media and processed for immunoblotting or immunofluorescence as above. For infection and foci experiments, taxol was added at 5 hpi and tubacin was added at 2.5 hpi. The concentrations and incubation time of taxol and tubacin were optimized based on previous reports[53,70–72].

**Split kinesin assay.** CV-1 cells expressing FKBP- and/or FRB-tagged proteins for at least 24 h were infected with SV40. Cells were treated at 0 hpi or 5 hpi with 1 μM of the rapa. Infection was allowed to proceed for total 24 hpi or 16 hpi for TAg expression or foci formation studies, respectively. Cells were then processed for immunoblotting or immunofluorescence as described above.

**Live-cell imaging.** For live-cell imaging, cells were seeded on 35 mm glass-bottom tissue culture dishes (Greiner Bio-one, Germany), triple transfected with GFPBAP31, FKBP-mChKIF5 DN and KIF5 motorBFP-FRB, and allowed to express the proteins for at least 24 h before infected with SV40 (MOI 20). Cells (16 hpi) were treated with the rapa linker. In total, 23 cells were imaged for the experimental condition (cells expressing GFPBAP31, FKBP-mChKIF5 DN and KIF5 motorFRB), whereas ten cells were imaged for the control condition (cells expressing GFPBAP31 and FKBP-mChKIF5 DN). Imaging of the cells was started at 0 h post-rapa addition for 2 h in cDMEM media without phenol red (Gibco). During imaging, cells were maintained at 5% CO$_2$ in a humidified chamber (Tokai Hit) regulated at 37 °C. The entire set-up was placed on a Nikon Ti-E/B microscope equipped with a × 100 1.49 numerical aperture oil-immersion objective warmed to 37 °C. Imaging was recorded using 20 mW diode lasers (488 nm) every 5 min for 2 h. ImageJ software (NIH) was used for image processing, analysis and assembly.

**Statistical analysis.** Data are represented as the average of the mean values from at least three independent experiments ($n \geq 3$) and the error bar represents s.d. Data are plotted using GraphPad Prism software, version 5.0b. Two-tailed Student's $t$-test was performed where indicated, to compare experimental data sets with the control. $*P < 0.05$, $**P < 0.01$, $***P < 0.001$ and $****P < 0.0001$ were considered to be significant, unless otherwise noted.

**Data availability.** All relevant data are available from the authors upon request.

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

## Acknowledgements

We thank members of Tsai Lab and Verhey Lab for discussions and suggestions throughout this work. This work was supported by NIH grants R01AI064296 and R01GM113722 to B.T. and R01GM070862 to K.J.V.

## Author contributions

M.S.R., M.F.E., K.J.V. and B.T. conceived and designed the experiments. M.S.R. and M.F.E. performed the experiments. M.S.R., M.F.E., K.J.V. and B.T. analysed the data. M.S.R., M.F.E., K.J.V. and B.T. contributed reagents/materials/analysis tools. M.S.R. and B.T. wrote the paper.

## Additional information

**Competing interests:** The authors declare no competing financial interests.

**Publisher's note**: 

