## [Peer Review File · Nature Communications]

Reviewers' comments:

Reviewer #1 (Remarks to the Author):

In this study, Ravindran et al. examine how SV40 virus penetrates the membrane to access the cytosol. Following up on an earlier observation that an ER membrane protein B14, that is important for this process, might interact with kinesin family proteins, the authors show that B14 interacts with Kif5 (kinesin-1). Using dominant negative and siRNA approaches authors highlight a role for Kif5 in SV40 replication and exit from the ER in particular. Further data is provided indicating that Kif5 is required for the formation/coalescence of SV40 foci. Using an elegant rapamycin mediated dimerisation approach, the authors show that the Kif5 dominant negative effect can be relieved by its association with the Kif5 motor domain, but not the motor domain from other kinesin family members. The authors argue that this is due to its preference for post-translationally modified MTs.

Overall, this is a nice study and the experiments appear to be well performed. The new role for Kif5 in supporting ER exit of SV40 is exciting, novel, and of broad interest, as is the potential insight that this relies on the preference of Kif5 for post-translationally modified MTs. However, I do think that some work is needed before it may become suitable for publication – in particular, some important controls and secondary validations of the principal findings are required.

Major Points

1. It is unclear how TAg expression was assessed in Figure 1B, D and elsewhere. Are cells simply highly positive or entirely negative for TAg, or was a cut-off used? If so, how was this defined? Representative images of TAg positive and negative cells should be provided. Similarly, it is unclear why western blot of TAg expression is not provided to support these conclusions throughout, as in Figure 1A.
2. In Figure 1E,F, the authors use siRNA to deplete Kif5B to support their conclusions using the Kif5 DN. It is important to exclude the possibility of off target effects from the siRNA, preferably by rescue, but if not, at a minimum, this should be validated using a secondary duplex. I feel that it is also important to confirm the author's observations elsewhere using this siRNA approach, particularly in relation to figure 4, where the morphology of foci is addressed for the first time, where it would seem important to exclude the possibility that expression of Kif5B DN has a global effect on the organization of the MT network.
3. Figure 5H is a bit superficial. How many cells is this movie representative of? These data should be supported with quantification. How often do foci fuse in non-treated cells? How much does this increase with rapamycin analogue treatment?
4. In Figure 6, no demonstration is provided that the FKBP-FRB mediated interaction of Kif17 and Kif1A (and Kif5B) works efficiently in this system. Ideally, it would be useful to know what proportion of the two proteins form a complex. Although other studies are cited (and I don't question the validity of the approach), given the importance of these data the

author's conclusions, this would seem to be crucial.

5. In Figure 7, the authors exclude the possibility that this requirement for kinesin-1 is due to its capacity to function under high load, but the connection to its preference for modified tubulin is very thin. Taxol has many effects on the MT network and I don't think this experiment is sufficient to support the author's conclusions regarding the use of the 'tubulin code', which is a central message of this manuscript. One alternative might be to inhibit HDAC6 either pharmacologically or by siRNA knockdown to test whether this enhances TAg expression or BAP31 foci formation.

6. In Figure 2, the authors should confirm that Kif5 DN can interact with B14.

Minor Points

Figure 2E : Model shows Kif5 tail penetrating the ER bilayer - this is likely misleading and should be amended.

Figure 2A-D : Whilst the conclusions appear sound, many of the western blots are over-exposed. These should be replaced with shorter exposures if possible.

Line 172: 'not much mCh' should be rephrased.

Reviewer #2 (Remarks to the Author):

The authors studied how Simian Virus 40 (SV40) penetrates the ER membrane to access the cytosol. They used a biochemical fishing approach based on immunoprecipitation from cell lysates containing overexpressed proteins of interest. Proteins of interest had earlier been implicated in SV40 infection by RNA interference screens. Based on these immunoprecipitation they construct a model that involves kinesin 1 and microtubules for SV40 infection. Some emphasis is placed on the idea where foci containing the transmembrane J protein B14 and BAP31 (as well as other factors, such as heat shock proteins) coalesce in the ER membrane through the action of kinesin 1 and specialized microtubules rich in acetylated tubulin. But the authors do not address the question if virus penetrates the membrane in these foci. There is also no direct interaction data, for example between B14 and kinesin 1. In general the study is at a rather high technical level and loosely supports the idea of a tubulin code, where specialized microtubules facilitate specific functions, for example in distinct subcellular regions.

Questions / issues

P6: Regarding time point of nocodazole addition which is critical to assess a function of microtubules post ER arrival of the viruses. The authors refer to some published work from others. It would be useful, however, to determine how many SV40 particles have in fact reached the ER 5-6h pi. This is relevant, since others observed that microtubules are

involved in SV40 transport from the plasma membrane to late endosomes and from there to the ER.

Fig 2B: How selective is this immunoprecipitation, in other words - do other Hsps coprecipitate?

Fig 2E: I do not think that the model is justified. There is no evidence here other than loosely defined immunocomplexes that would suggest that kinesin directly binds to B14, for example. This model should be removed in the opinion of this reviewer.

Fig 3, line 219: I am not sure if this conclusion is justified. There is really no evidence in this study that the penetration of SV40 occurs from the ER. Evidence for this is taken from the literature, and the literature is not always correct (see ref 12 for an example). It is formally possible that kinesin supports the cytosolic entry of SV40 from another compartment than the ER.

Minor points:

Line 62, ref 12: this is outdated and should no longer be cited, as there are not caveosomes in cells. For a reference, see {Engel, 2011, 21345959}.

Line 99: I cannot see how ref 24 refers to kinesin. Please refine statement, or refer to other literature.

Line 142: How broad is STLC? It was shown to inhibit KIF11 but is it a general Kif-5 inhibitor?

PLEASE indicate the apparent molecular weights on the gels and western blots throughout the manuscript.

Line 196: 'any' is a too strong statement. How many internal membranes were checked?

Reviewer #3 (Remarks to the Author):

Comment to the authors

In this manuscript Ravindran et al. describe a novel role for the major kinesin-1 motor protein in the ER-to-cytosol transport of the non-enveloped polyoma virus SV40. The authors show that the microtubule network is necessary for SV40 infectivity. They identify that kinesin-1 facilitates SV40 ER-to-cytosol translocation and establish its interaction with B14, a protein, which is recruited into SV40 induced foci assumed to be functional cytosol translocation sites for the virus. Furthermore the authors show that infectivity (measured by Tag expression) and foci formation specifically requires functional kinesin-1 motors.

This work establishes kinesin-1 as a novel molecular actor for the membrane penetration step of SV40, a non-enveloped virus, which will advance our understanding of this ill-defined process. The manuscript describing the involvement of kinesin-1 in SV40 infectivity and in generating an ER-to-cytosol penetration site (depicted as large focal structures and/or TAG expression) convinces with an elegant experimental approach and clear and logical data presentation. In contrast the relation of kinesin-1 function to the post-translational state of microtubules and the formation of the focal structures for infectivity is less conclusive and should be clarified.

Major points

- This study has two major readouts. Viral infectivity measured as T-antigen positive cells and the formation of large vs. small Bap31/Vp1 positive foci as functional ER-to-Cytosol transition sites. Unfortunately all assays for infectivity were done at an MOI of 0.5 while all assays for foci formation were done at an MOI of 20. The authors did not explain the rationale for using different MOIs in the text. Moreover in several text passages the authors link both events suggesting that foci formation is required for infectivity (e.g. lane 299-300, 386-388). The question I have is if foci formation under the different assay conditions also occurs at low MOI or if differences in infectivity (measured by Tag expression) are still observable at high MOI when differences in foci formations become apparent?
- Data from several figures suggest that a single large foci is formed in cells infected at high MOI which appear to localize to the perinuclear region reminiscent of the MTOC position (Fig. 4, Fig. 5 and Fig. 7, but not in Movie S1). This is especially visible in Fig.7 where localization of foci and convergence of microtubules seem to coincide. A priori this would not be the expected location for Kinesin-1 motor driven cargo as suggested in this study. The authors should first evaluate the localization of the foci in respect to the MTOC using pericentrin and/or gamma-tubulin stain vs. a possible Golgi or arbitrary localization. If foci localization coincides with MTOC localization the authors should re-evaluate their model.

Minor points

- Fig. 1B, E/F: The authors should show the specific inhibitory effect of KIF5 by also depleting KIF11 as a control, which should not affect the SV40 infectivity. This would also substantiate the data in Fig. B
- All examples showing cells expressing the DN form of KIF5 (tail domain) appear to show bundled microtubules (Fig. 4A/C, Fig.5E). The authors explain this as a common feature of the construct and cite Ref.34. However, in the cited reference only MT binding of the tail domain is shown and not MT bundling. Could this be an effect of the fusion-protein? Would siRNA knockdown of KIF5 be an alternative?
- normalizations in several graphs should not involve % but rather arbitrary units (e.g. Fig. 7E, where the legend suggests that 200% (!) of cells following Taxol treatment have BAP31 foci).
- Please show a loading control for Fig. 1C/5B
- The authors suggest that kinesin-1 promotes SV40 infectivity through its preference for acetylated MT (e.g. lane 370-371). No data in support of this claim are shown. The authors should consider to validate this by using split motors (e.g. KIF5 motor domain vs. KIF17/1A motor domain, as in Fig. 6) in the presence of increasing amounts of Taxol (as in Fig.7F) and show specificity for the KIF5 motor domain in promoting viral infectivity. Otherwise the authors should moderate their claim throughout the text and abstract.

Reviewer #1:

In this study, Ravindran et al. examine how SV40 virus penetrates the membrane to access the cytosol. Following up on an earlier observation that an ER membrane protein B14, that is important for this process, might interact with kinesin family proteins, the authors show that B14 interacts with Kif5 (kinesin-1). Using dominant negative and siRNA approaches authors highlight a role for Kif5 in SV40 replication and exit from the ER in particular. Further data is provided indicating that Kif5 is required for the formation/coalescence of SV40 foci. Using an elegant rapamycin mediated dimerization approach, the authors show that the Kif5 dominant negative effect can be relieved by its association with the Kif5 motor domain, but not the motor domain from other kinesin family members. The authors argue that this is due to its preference for post-translationally modified MTs.

Overall, this is a nice study and the experiments appear to be well performed. The new role for Kif5 in supporting ER exit of SV40 is exciting, novel, and of broad interest, as is the potential insight that this relies on the preference of Kif5 for post-translationally modified MTs. However, I do think that some work is needed before it may become suitable for publication – in particular, some important controls and secondary validations of the principal findings are required.

Major Points

1. It is unclear how TAg expression was assessed in Figure 1B, D and elsewhere. Are cells simply highly positive or entirely negative for TAg, or was a cut-off used? If so, how was this defined? Representative images of TAg positive and negative cells should be provided.

When CV-1 cells were infected with SV40 and stained for the expression of TAg, the infected (TAg-positive) cells were clearly distinguishable from the uninfected cells, as can be seen in the representative images presented in Figure 1B. Similar to our previous publications, the extent of TAg expression was manually quantified by the ImageJ software program using cell counter plugin. A detailed analysis of SV40 infection is now included in the Methods section (lines 664-682).

Similarly, it is unclear why western blot of TAg expression is not provided to support these conclusions throughout, as in Figure 1A.

As requested, we now provide TAg immunoblots for siRNA-mediated knockdown (Figure 1G) or drug-treated (Figure 1C and 7G/S4E) infection experiments performed in CV-1 cells. We did not include TAg immunoblots for infection experiments using the dominant-negative overexpression approach. This is because the low transfection efficiency of CV-1 cells (10-20%) precludes us from observing any significant phenotype in the TAg level when using a whole cell lysate derived from pooled cells. As such, for these dominant-negative experiments, infection can only be analyzed by scoring for the extent of TAg expression in those cells that are expressing the dominant-negative (or control) constructs by immunofluorescence.

2. In Figure 1E, 1F, the authors use siRNA to deplete Kif5B to support their conclusions using the Kif5 DN. It is important to exclude the possibility of off-target effects from the siRNA, preferably by rescue, but if not, at a minimum, this should be validated using a secondary duplex.

As requested, we have now used another siRNA (KIF5B^{#2} siRNA) to support our claim that KIF5B knockdown blocks SV40 infection (Figure 1G/1H). Moreover, we have performed a rescue experiment in which expression of the KIF5 family member KIF5C largely restored the block in SV40 infection due to KIF5B knockdown (Figure 1I), unambiguously demonstrating that the effect of the KIF5B siRNA is not due to off-target effects. We have also provided the experimental details in the Methods section (line 604-616).

I feel that it is also important to confirm the author's observations elsewhere using this siRNA approach, particularly in relation to figure 4, where the morphology of foci is addressed for the first time, where it would seem important to exclude the possibility that expression of Kif5B DN has a global effect on the organization of the MT network.

We agree with the concern raised by this reviewer. As requested, we have now analyzed BAP31 foci formation in KIF5B knockdown CV-1 cells (Figure 4F), with a set of representative images provided in Figure 4E. Our results demonstrate that KIF5B

depletion reduced large (and increased small) foci formation, consistent with findings using the dominant-negative overexpression approach (Figure 4A/4B).

3. Figure 5H is a bit superficial. How many cells is this movie representative of? These data should be supported with quantification. How often do foci fuse in non-treated cells? How much does this increase with rapamycin analogue treatment?

We apologize for not explaining the live-cell imaging experiments more clearly. Figure 5H and the supplementary movie represent a total 33 cells (23 experimental and 10 negative control). Under the experimental condition (cells expressing GFP^{BAP31} , $FKBP^{KIF5\ DN}$, and $KIF5\ motor^{FRB}$) where 23 cells were imaged, we observed fusion events leading to formation of the single large BAP31 focus in 17 cells (74%) after addition of the A/C linker. By contrast, for the negative control (cells expressing GFP^{BAP31} and $FKBP-mCh^{KIF5\ DN}$) where 10 cells were imaged and we observed fusion events in only 1 cell (10%) after addition of the A/C linker. Thus, there is a greater than 7-fold difference between the experimental and control conditions. We now state these quantifications in the Results section (line 347-354), and provided additional details in the Methods section (line 714-728).

4. In Figure 6, no demonstration is provided that the FKBP-FRB mediated interaction of Kif17 and Kif1A (and Kif5B) works efficiently in this system. Ideally, it would be useful to know what proportion of the two proteins form a complex. Although other studies are cited (and I don't question the validity of the approach), given the importance of these data the author's conclusions, this would seem to be crucial.

To address this point, we have now performed co-immunoprecipitation experiments, and demonstrate that only in the presence of the A/C heterodimerizer did immunoprecipitation of motor-less KIF5 ($FKBP-mCh^{KIF5\ DN}$) significantly pull down the corresponding motor domain partners $KIF5\ motor^{FRB}$, $KIF17\ motor^{FRB}$, and $KIF1A\ motor^{FRB}$ (Supplementary Figure 2). These findings validate the integrity of the FKBP-FRB dimerization system, in agreement with published work (Kapitein et al 2010 Biophys J, Robinson et al 2010 Dev Cell, Jenkins et al 2012 JCB, van Spronsen et al 2013 Neuron, Kapitein et al 2013 Curr Biol, Bentley et al 2015 JCB). We have now cited some of these articles in the Results

section (line 636-643).

5. In Figure 7, the authors exclude the possibility that this requirement for kinesin-1 is due to its capacity to function under high load, but the connection to its preference for modified tubulin is very thin. Taxol has many effects on the MT network and I don't think this experiment is sufficient to support the author's conclusions regarding the use of the 'tubulin code', which is a central message of this manuscript. One alternative might be to inhibit HDAC6 either pharmacologically or by siRNA knockdown to test whether this enhances TAg expression or BAP31 foci formation.

As requested, we used the HDAC6 inhibitor tubacin, and found that cells treated with this drug increased SV40 infection (Supplementary Figure 4E/4F) and foci formation (Supplementary Figure 4G), consistent with results using taxol (Figure 7). A representative image of acetylated-tubulin in tubacin-treated cells is also provided in Supplementary Figure 4D. We have also stated this in the Results section (line 430-437), and provided additional details in the Methods section (line 699-705).

6. In Figure 2, the authors should confirm that Kif5 DN can interact with B14.

As requested, we now include new data in Figure 2F, demonstrating that immunoprecipitation of endogenous B14 co-immunoprecipitates transfected ^{mCh}KIF5 DN. This finding demonstrates that KIF5 DN can interact with B14.

Minor points

Figure 2E: Model shows Kif5 tail penetrating the ER bilayer - this is likely misleading and should be amended.

Because this reviewer and reviewer #2 (comment 3) raised the same concern, we have decided to remove this model.

Figure 2A-D: Whilst the conclusions appear sound, many of the western blots are over-exposed. These should be replaced with shorter exposures if possible.

As requested, we have now replaced some over-exposed blots with less-exposed blots (Figure 2A, Hsc70 input blot; Figure 2D, Hsp105 input blot), or replaced with entirely new

experimental data (Figure 2E).

Line 172: 'not much mCh' should be rephrased.

We have changed the statement accordingly.

Reviewer #2

The authors studied how Simian Virus 40 (SV40) penetrates the ER membrane to access the cytosol. They used a biochemical fishing approach based on immunoprecipitation from cell lysates containing overexpressed proteins of interest. Proteins of interest had earlier been implicated in SV40 infection by RNA interference screens. Based on these immunoprecipitation they construct a model that involves kinesin 1 and microtubules for SV40 infection. Some emphasis is placed on the idea where foci containing the transmembrane J protein B14 and BAP31 (as well as other factors, such as heat shock proteins) coalesce in the ER membrane through the action of kinesin 1 and specialized microtubules rich in acetylated tubulin. But the authors do not address the question if virus penetrates the membrane in these foci. There is also no direct interaction data, for example between B14 and kinesin 1. In general the study is at a rather high technical level and loosely supports the idea of a tubulin code, where specialized microtubules facilitate specific functions, for example in distinct subcellular regions.

Questions / issues

P6: Regarding time point of nocodazole addition which is critical to assess a function of microtubules post ER arrival of the viruses. The authors refer to some published work from others. It would be useful, however, to determine how many SV40 particles have in fact reached the ER 5-6hpi. This is relevant, since others observed that microtubules are involved in SV40 transport from the plasma membrane to late endosomes and from there to the ER.

As requested, we have now performed an experiment using the cell-based ER-arrival assay established by Inoue et al, (PLoS Path., 2011). In this assay, SV40 ER arrival from the cell surface was analyzed by appearance of VP1 monomers and dimers, which can be observed by non-reducing SDS-PAGE. This is because SV40's disulfide bonds are reduced and isomerized when the virus successfully reaches the ER from the cell surface, leading to formation of VP1 monomers and dimers. Thus, under the non-reducing condition, generation of VP1 monomers/dimers reflects virus arrival to the ER. Using this approach, we found that addition of nocodazole at 0 hpi decreases VP1 monomer (and dimer) formation (Supplementary Figure 1), indicating that SV40 ER arrival is impaired. This suggests that cell surface-to-ER transport of SV40 requires intact microtubules, consistent with several previous reports. However, addition of

nocodazole at 5 hpi does not significantly decrease VP1 monomer (and dimer) formation, suggesting that the majority of the virus that can reach the ER from the cell surface has already arrived in the ER at 5 hpi. Thus, when nocodazole is added at this time point, any effect this drug has on foci formation or infection is likely due to a post ER-arrival effect.

Fig 2B: How selective is this immunoprecipitation, in other words - do other HSPs co-precipitate?

As requested, we now demonstrate that immunoprecipitation of transfected ^{FLAG}B14 co-precipitates endogenous Hsc70, Hsp105, and SGTA (as expected), but not Hsp27, HspB5, and HspBP1 (Figure 2C). Thus, B14 displays selectivity in the HSPs that it engages.

Fig 2E: I do not think that the model is justified. There is no evidence here other than loosely defined immuno-complexes that would suggest that kinesin directly binds to B14, for example. This model should be removed in the opinion of this reviewer.

As this reviewer and reviewer #1 raised a concern about this model, we have decided to remove it.

Fig 3, line 219: I am not sure if this conclusion is justified. There is really no evidence in this study that the penetration of SV40 occurs from the ER. Evidence for this is taken from the literature, and the literature is not always correct (see ref 12 for an example). It is formally possible that kinesin supports the cytosolic entry of SV40 from another compartment than the ER.

The best evidence that cytosol entry of SV40 is from the ER is based on numerous studies demonstrating that specific depletion of ER-resident (luminal or membrane) proteins block arrival of the virus to the cytosol, and consequently, virus infection. Our laboratory (Bagchi P. et al 2016; Ravindran MS, 2015; Inoue T, 2015; Walczak CP, 2014; Inoue T, 2009), the Helenius laboratory (Geiger R, 2011; Schelhaas M, 2007), the Norkin laboratory (Kuksin D, 2012) and the Dimaio laboratory (Goodwin EC, 2011), have all reported on this and our data are largely consistent. Moreover, studies with other polyomaviruses, including the human BK and JC viruses (Jiang M, 2009; Nelson C, 2012), as well as the murine polyomavirus (Qian M, 2009; Rainey-Barger EK, 2007), have also shown that selectively disrupting ER-resident factors impair virus infection.

Because most of these primary references were included in a Nature Reviews Microbiology article that my laboratory recently wrote (Ravindran MS, 2016), we have now cited this review in the Discussion section.

Minor points

Line 62, ref 12: this is outdated and should no longer be cited, as there are not caveosomes in cells. For a reference, see (Engel, 2011, 21345959).

We have now removed reference 12 from the manuscript.

Line 99: I cannot see how ref 24 refers to kinesin. Please refine statement, or refer to other literature.

We have replaced ref 24 in line 99 with the Strunze S et al. reference #26.

Line 142: How broad is STLC? It was shown to inhibit KIF11 but is it a general Kif-5 inhibitor?

STLC is an inhibitor that is specific to kinesin motors in the kinesin-5 family. It does not inhibit kinesin-1, kinesin-6, kinesin-7, kinesin-10, kinesin-13, or kinesin-14 motors in *in vitro* assays (Skoufias et al, 2006 JBC) because this inhibitor binds to a unique loop on the surface of the kinesin-5 motor (Brier et al, 2004 Biochem). Moreover, STLC is specific for mammalian members of the kinesin-5 family and does not inhibit the *Drosophila* kinesin-5 homologue (Klp61F) (Liu et al, 2011, JBC). KIF11 is the only kinesin-5 family member in mammalian cells (Wickstead 2010, BMC Evol Biol). We have now cited some of these articles in the Results section (line 137).

PLEASE indicate the apparent molecular weights on the gels and western blots throughout the manuscript.

We have now included protein molecular weights throughout the figures.

Line 196: 'any' is a too strong statement. How many internal membranes were checked?

We have changed this statement and it now reads as: "permeabilizes the plasma membrane without damaging the ER membrane".

Reviewer #3

In this manuscript Ravindran et al. describe a novel role for the major kinesin-1 motor protein in the ER-to-cytosol transport of the non-enveloped polyomavirus SV40. The authors show that the microtubule network is necessary for SV40 infectivity. They identify that kinesin-1 facilitates SV40 ER-to-cytosol translocation and establish its interaction with B14, a protein, which is recruited into SV40 induced foci assumed to be functional cytosol translocation sites for the virus. Furthermore the authors show that infectivity (measured by TAg expression) and foci formation specifically requires functional kinesin-1 motors.

This work establishes kinesin-1 as a novel molecular actor for the membrane penetration step of SV40, a non-enveloped virus, which will advance our understanding of this ill-defined process. The manuscript describing the involvement of kinesin-1 in SV40 infectivity and in generating an ER-to-cytosol penetration site (depicted as large focal structures and/or TAg expression) convinces with an elegant experimental approach and clear and logical data presentation. In contrast the relation of kinesin-1 function to the post-translational state of microtubules and the formation of the focal structures for infectivity is less conclusive and should be clarified.

Major points

1. This study has two major readouts. Viral infectivity measured as T-antigen positive cells and the formation of large vs. small BAP31/VP1 positive foci as functional ER-to-Cytosol transition sites. Unfortunately all assays for infectivity were done at an MOI of 0.5 while all assays for foci formation were done at an MOI of 20. The authors did not explain the rationale for using different MOIs in the text. Moreover in several text passages the authors link both events suggesting that foci formation is required for infectivity (e.g. line 299-300, 386-388). The question I have is if foci formation under the different assay conditions also occurs at low MOI or if differences in infectivity (measured by TAg expression) are still observable at high MOI when differences in foci formations become apparent?

As requested, we have now performed the infection experiments (Figure 1F) at three different MOI (0.5, 5, and 20), and the foci formation experiments (Figure 4B) at two different MOI (5 and 20). Importantly, our results demonstrate that expression of dominant-negative KIF5 (KIF5 DN) reduced infection and blocked large foci formation

regardless of the MOI. We did not use the lowest MOI (0.5) for the foci experiments because foci formation is difficult to observe by immunofluorescence at this virus concentration.

2. Data from several figures suggest that a single large foci is formed in cells infected at high MOI which appear to localize to the perinuclear region reminiscent of the MTOC position (Fig. 4, Fig. 5 and Fig. 7, but not in Movie S1). This is especially visible in Fig.7 where localization of foci and convergence of microtubules seem to coincide. A priori this would not be the expected location for Kinesin-1 motor driven cargo as suggested in this study. The authors should first evaluate the localization of the foci in respect to the MTOC using pericentrin and/or gamma-tubulin stain vs. a possible Golgi or arbitrary localization. If foci localization coincides with MTOC localization the authors should re-evaluate their model.

As requested, we now demonstrate in Supplementary Figure 3 that the large SV40-induced (BAP31-positive) focus does not co-localize with MTOC (γ -tubulin), Golgi (Giantin), or the early endosomes (EEA1).

Minor points

3. Fig. 1B, E/F: The authors should show the specific inhibitory effect of KIF5 by also depleting KIF11 as a control, which should not affect the SV40 infectivity. This would also substantiate the data in Fig. B.

We thank this reviewer for the suggestion. Indeed, we attempted to knock down KIF11 in CV-1 cells using a KIF11-specific siRNA (custom synthesized, Invitrogen). However, all of the cells died within a day after transfection of the siRNA. This observation is not surprising since KIF11 knockdown is often used to arrest cells in mitosis and prolonged mitotic arrest leads to apoptosis (see discussion in Wojcik et al 2013 Gene). In fact, this finding illustrates the major advantage of using the (KIF11 specific) STLC inhibitor (Figure 1C/1D) because it enables an investigator to observe the phenotype of KIF11 inhibition upon acute inhibition.

4. All examples showing cells expressing the DN form of KIF5 (tail domain) appear to show bundled microtubules (Fig. 4A/C, Fig.5E). The authors explain this as a common feature of the construct and cite Ref.34. However, in the cited reference only MT binding of the tail domain is shown and not MT bundling. Could this be an effect of the

fusion-protein? Would siRNA knockdown of KIF5 be an alternative?

That the expressed KIF5 tail domain binds to microtubules is known in the kinesin field (e.g. Navone et al 1992 JCB; Hackney & Stock 2000 NCB; Straube et al 2006 MBoC; Yonekura et al 2006 BBRC; Seeger & Rice 2010 JBC) and because of this, most people who use this construct as a dominant negative do not show images of the transfected cells (e.g. Wubbolts et al 1999 JCS; Setou et al 2002 Nature; Ravikumar et al 2005 Nat Gen; van Spronsen et al 2013 Neuron). Whether the KIF5 tail binds along microtubules (as in the images shown in Navone et al 1992 JCB) or bundles microtubules (as in the images shown in Figure 4) depend on the level of expression, the length of time of expression, and the cell type. We have now cited some of these articles in the Results section (line 262).

As requested, we now used a second siRNAs to knockdown KIF5B, and demonstrate that SV40 infection is robustly impaired under both knockdown conditions (Figure 1G and 1H). In addition, we have performed a rescue experiment in which we demonstrate that the block in SV40 infection when KIF5B is depleted can be restored by the expression of the KIF5 family member KIF5C (Figure 1I). Moreover, this siRNA knockdown strategy was used to assess foci formation. Consistent with the dominant-negative approach, we now find that depleting KIF5B also decreases large (and increases small) foci formation (Figure 4G), with a representative image provided in Figure 4F. Collectively, these data strengthen the importance of KIF5 in controlling virus-induced foci formation and SV40 infection. We have also provided the experimental procedure in the Methods section (line 604-616).

5. Normalizations in several graphs should not involve % but rather arbitrary units (e.g. Fig. 7E, where the legend suggests that 200% (!) of cells following Taxol treatment have BAP31 foci).

We have now changed the label of the Y-axis throughout the manuscript. The Y-axis now appears either as: TAg expression (normalized to control) or BAP31 foci (normalized to control).

6. Please show a loading control for Fig. 1C/5B.

We have now provided Hsp90 as a loading control for Figure 1E (formerly 1C) and

Figure 5B, as requested.

7. The authors suggest that kinesin-1 promotes SV40 infectivity through its preference for acetylated MT (e.g. lane 370-371). No data in support of this claim are shown. The authors should consider to validate this by using split motors (e.g. KIF5 motor domain vs. KIF17/1A motor domain, as in Fig. 6) in the presence of increasing amounts of Taxol (as in Fig.7F) and show specificity for the KIF5 motor domain in promoting viral infectivity. Otherwise the authors should moderate their claim throughout the text and abstract.

As requested, we have performed the split motor experiment in the presence of taxol, as shown in Supplementary Figure 4. We found that expressing KIF5 DN potently blocks SV40 infection in the presence of taxol. Additionally, in the presence of taxol, addition of the A/C heterodimerizer rescues SV40 infection only in cells co-expressing KIF5 DN and the KIF5 (but not KIF17 or KIF1A) motor, demonstrating that the specific requirement for kinesin-1 (but not kinesin-2 and kinesin-3) during SV40 infection is maintained in the taxol-treated condition. These findings are consistent with our observations in Figures 6 and 7, strengthening our claim that SV40 selects kinesin-1 due to this motor's preference for acetylated microtubules.

Furthermore, we now provide new data demonstrating that the HDAC6 (tubulin deacetylase) inhibitor tubacin caused an increase in both SV40 infection (Supplementary Figure 4E/4F) and virus-induced foci formation (Supplementary Figure 4G), consistent with results using taxol (Figure 7). These results further support the importance of acetylated microtubules in SV40 infection and foci formation.

We note that when this article was under revision, Hornikova et al, reported that when cells are transiently transfected with the murine polyomavirus VP1 protein, microtubules became hyper-acetylated. This finding is in agreement with our hypothesis that SV40 co-opts kinesin-1, which preferentially uses acetylated microtubules, during host entry.

REVIEWERS' COMMENTS:

Reviewer #1 (Remarks to the Author):

The authors have done a good job of addressing my concerns in the revised manuscript, which I now believe is suitable for publication in Nature Communications.

Reviewer #2 (Remarks to the Author):

the authors have adequately addressed the points of my concern.

Reviewer #3 (Remarks to the Author):

The authors have fully responded to my remarks